# The plague of 1720 and migration in Martigues (France) in the 17th and 18th centuries

Pierre Darlu[1]☺*, Isabelle Séguy[2]☺

1 UMR7206 Eco-anthropologie, CNRS, MNHN, Musée de l'Homme, Paris, France, 2 Institut national d'études démographiques (French Institute for Demographic Studies, INED), Paris, and Université Côte d'Azur, CNRS, UMR7264 CEPAM, Nice, France

☺ These authors contributed equally to this work.
* pierre.darlu@mnhn.fr

## Abstract

One-off events such as wars or epidemics can change the structure of populations, by encouraging the mobility or causing the death of certain categories of people. They can also lead to a demographic slump, made up for by the arrival of new migrants. The town of Martigues (Bouches-du-Rhône, France) provides an example of this kind of local population renewal. Between the seventeenth and eighteenth centuries, its population was partially renewed after the plague of 1720. To measure the effect of the plague as a disruptive event, data on the surnames of people born in Martigues before and after the epidemic of 1720 were collected and analysed. After a delicate stage of lemmatization of the surnames, three categories of names were distinguished: those present before 1720 and which disappeared from Martigues after the plague; those absent before 1720 and which appeared afterwards; and surnames that were continually present in Martigues, but whose frequency in terms of the number of births per year could be contrasted between before and after 1720. The surname data for these three categories is compared with the list of names of the victims of the plague, which makes it possible to envisage a possible reason for their disappearance. In addition, the frequencies of surnames in the 18th century in the Bouches-du-Rhône department and in France in the 19th century make it possible to locate the possible destination of people whose surnames disappeared after 1720, as well as the probable origin of the surnames of people arriving after 1720. This study helps us to understand how the impact of an epidemic crisis can affect the evolution of a population on a local scale. The surname method used here indicates that the plague caused an exceptionally large renewal of approximately 50% of the stock of surnames, and thus of the population bearing those names. It also shows that fertility declined significantly among individuals whose surnames were already present among those who died of the plague. Finally, the results demonstrate that population renewal was achieved primarily through immigration, mainly from neighbouring municipalities.

**Data availability statement:** All relevant data are within the paper and its Supporting information files.

**Funding:** The author(s) received no specific funding for this work.

**Competing interests:** The authors have declared that no competing interests exist.

## 1. Introduction

The objective of this study is to illustrate, based on an example, the effects of events such as wars or epidemics on the structure of a population, leading to an increase in deaths and the exile of a part of the population, and resulting in a possible slump in the population, offset by the arrival of new migrants.

The example presented here focuses specifically on the local demographic consequences of the last major plague epidemic in France, during the early modern period, using an innovative surname-based approach. This method is suitable because, by the eighteenth century, surnames were already passed down from father to children, albeit sometimes with minor variations in spelling. No surname could appear spontaneously, as every individual bore a baptismal name, even those who came from outside the community. Consequently, analyzing changes in a population's stock of surnames provides a reliable indicator of either extinction, disappearance, or migration. Previous studies have demonstrated the effectiveness of this method in various contexts and periods, both past and present [1–10].

The plague epidemic of 1720–1722 began when, on 25 May 1720, an infected ship from the eastern Mediterranean arrived in Marseille. Due to a series of administrative failures, the contagion spread from the infirmaries throughout the city, taking its first recorded victim on 20 June 1720 [11–13]. The outbreak expanded rapidly. Despite quarantines—undoubtedly implemented too late—it gained momentum in the surrounding villages, spreading across Provence and into Lower Languedoc and the Comtat Venaissin. The epidemic's toll varied sharply from one community to another: some were spared completely, while others suffered severe losses, such as La Valette, with 1,068 deaths out of 1,600 inhabitants [12]. Urban centres were the most severely affected. Overall, between June 1720 and October 1722, the epidemic struck 146 localities, resulting in approximately 120,800 deaths in an estimated population of 400,000 [14].

Caused by the bacterium *Yersinia pestis* and extensively documented through paleogenomic studies [15–18], the plague exhibits very distinctive epidemiological characteristics. It progresses rapidly among exposed individuals and is highly lethal regardless of age or gender. Consequently, although the population size was drastically reduced by 1722, its structure remained largely stable in the immediate aftermath of the epidemic [19]. In the medium term, however, outcomes differed depending on both the resilience (or lack thereof) of local populations and the effectiveness of community policies to attract and retain new inhabitants.

To assess the epidemic-related dynamics of a historical population, this study draws on the surnames of inhabitants of a small town, Martigues (Fig 1), including those who lived there during the epidemic and those appearing in parish registers over the following half-century. Such an approach is rarely possible, as it requires uninterrupted parish records and, even more unusually, a complete list of the epidemic's victims.

The Ordinance of Saint-Germain-en-Laye (1667) required the preservation of duplicate parish registers and standardized the content to be recorded for baptisms, marriages, and burials across the kingdom. As a result, parish series—often

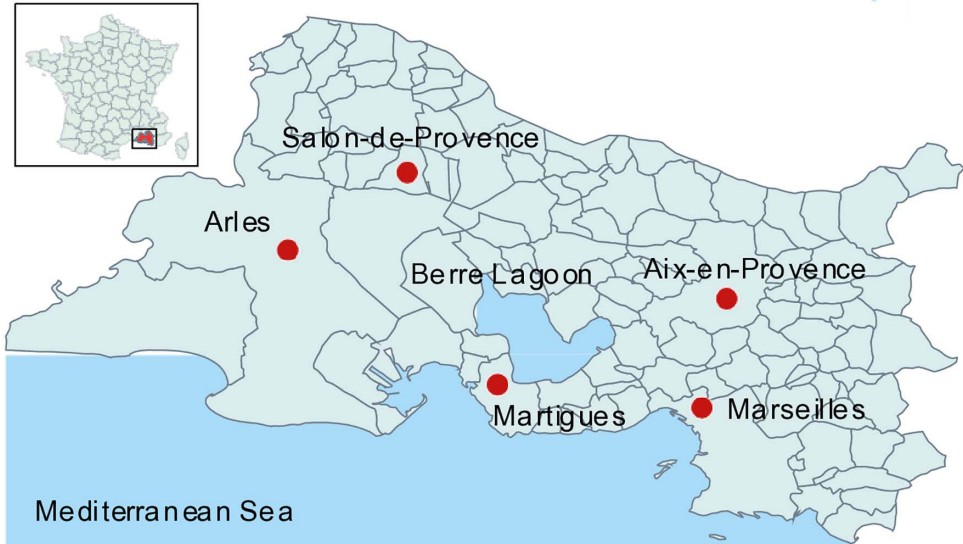

**Fig 1. Location of Martigues.** Martigues is a city located in Bouches-du-Rhône department, France, between Berre Lagoon and Mediterranean Sea. See also the location of Marseilles and other cities. Original map created by the authors using the free software *Philcarto* (http://philcarto.free.fr), with base map © IGN – AdminExpress.

extending further back in time—are generally complete from the late seventeenth century onward, except for losses caused by historical events. Lists of plague victims, by contrast, are far rarer: the epidemic disrupted community life, interrupted vital registration, and even when retrospective lists were drawn up to settle inheritance claims, they have seldom survived. In rare cases, communities in Provence have partially preserved records identifying the victims of the 1720–1722 epidemic. Martigues stands out for its exceptionally rich documentation covering both the medieval and early modern periods, enabling close study of the context in which the 1720 plague epidemic unfolded.

The first victim of the plague was recorded on 6 October 1720 [20] but Martigues was not officially declared as contaminated before 9 November 1720, and the town would remain so until 17 June 1721. Between these two dates, historical sources point to 2,150 victims [21] out of a population of 5,886 officially counted in 1716 (S1.1 in S1 File).

The parish registers, conserved in the municipal archives of Martigues (S1.2 in S1 File), recorded the deaths in the customary manner at the start of the epidemic and subsequently in the form of a list drawn up in the wake of the contagion. These documents provide the surname, given name, sex and age of the victims, as well as the date and place of death, the identity of the father and mother or spouse, the occupation, and, in some cases, kinship information. It was not possible to retrospectively provide a name for all the plague victims. For some, we know the surname and the given name, while others remain entirely anonymous. Previous research [22,23] has produced a list of the plague victims in Martigues based on death certificates. This figure is close to that of the Martigues authorities at the end of the epidemic.

From a socio-economic standpoint, the town of Martigues was dominated by seafaring occupations, with ship captains, captains of more modest vessels, and sailors accounting for 60% of the family heads subject to the capitation tax in 1702 (S1.3 in S1 File). Merchants, traders, craftsmen and tradespeople accounted for 20% of the town's population; farmer owners, day labourers, mule drivers, land workers and quarry workers represented 15% of taxpayers and lived on the land, away from the town itself. The wealthiest class (members of the nobility, learned professions, doctors, surgeons and apothecaries, men of the law, officers and members of the clergy) accounted for just 5% of taxpayers [24].

The key questions here are determining whether the epidemic impacted some individuals more than others [19], and whether the people no longer attested to in Martigues after 1720 were victims of the plague, survived it, or migrated elsewhere.

Did the families having survived the plague, and who by doing so ensured a certain continuity in local genealogy, experience an increase in the number of children after the plague comparable with that before the plague? Was the town repopulated through the immigration of people from neighbouring municipalities, Lower Provence or even further afield, and in what proportion? To report on these population movements, the method used here is based on an observation of the changes having occurred in the surname corpora in the pre-plague period (1689–1720, period P1) and the post-plague period (1721–1789, period P2). We will take account of the fact that Martigues at the time was composed of four districts, La Couronne, Ferrière, L'Île and Jonquières.

## 2. The data

### 2.1. The parish registers of Martigues and its districts, provided by Geneanet

The list of baptisms is drawn from the parish registers in the municipal archives of Martigues(S1.1 in S1 File). Registers begin in 1539 in L'Île (with a gap from 1581 to 1601), in 1635 in Jonquières (gap: 1638–1640), and in 1668 in Ferrières, while La Couronne—a subsidiary parish of Jonquières—has registers dating from the first third of the eighteenth century. Apart from the gaps mentioned above, the series is continuous until 1792. From the early eighteenth century onwards, the entries become increasingly complete and detailed.

All baptism, marriage, and burial records (BMS) from 1701 to 1725 were transcribed in our previous research [19,24]. The present study, however, relies solely on a secondary source: genealogists' transcriptions, beginning in 1689. These were compared with our own partial transcriptions, and the genealogists' datasets were preferred for their reliability and broader chronological coverage, even though their full transcripts were not yet entirely available at the time of this study.

The data from this secondary source are composed of baptism certificates collected by the genealogical association of Bouches-du-Rhône for 1689–1789 and provided to us by Geneanet (referred to hereafter as the "GᴇɴB-ᴍᴀʀ" file). These files contain the names of the baptized, the date of their baptism, the names of their father and mother, and, often, the father's profession. This represents 29,830 baptisms. We have excluded from the initial Geneanet database certain records for which the surname was not usable (ᴜɴᴋɴᴏᴡɴ, X, F,..., N,?, etc.). All the names were transformed into capitals to avoid any diacritic signs that can prove extremely variable when present and to enable comparisons with other databases in which the names are exclusively in capitals. We have conserved the records from which the name of the father and/or mother of the baptized are missing. The total number of the different surnames retained for the entire 1689–1789 period is 4,022 (see Table 1). We will make no distinction between baptism certificate and birth certificate, since the main point

Table 1. Distribution of the number of different surnames according to the number N of births Classes of N are from 0 to more than 10 births by name, in Martigues before and after 1720, and before lemmatization. The S's are marginal numbers (e.g., *38* different names are carried by 1 baptized person before 1720 and by 2 after).

| | | After 1720 | | | | | | |
|---|---|---|---|---|---|---|---|---|
| | N | 0 | 1 | 2 | 3 | 4–10 | > 10 | S |
| Before 1721 | 0 | | 986 | 222 | 80 | 164 | 42 | 1,494 |
| | 1 | 1223 | 93 | *38* | 17 | 55 | 17 | 1,443 |
| | 2 | 229 | 29 | 23 | 13 | 28 | 19 | 341 |
| | 3 | 96 | 25 | 9 | 10 | 20 | 15 | 175 |
| | 4–10 | 122 | 38 | 28 | 16 | 78 | 83 | 365 |
| | > 10 | 10 | 7 | 7 | 2 | 25 | 153 | 204 |
| | S | 1,680 | 1,178 | 327 | 138 | 370 | 329 | 4,022 |

is the names of the baptized individuals and not strictly speaking those of births, even though at the time almost all births were followed in the next 24 hours by baptisms.

We could also have used the register of deaths. But while the place of death is Martigues, nothing permits us to determine the place of birth of the deceased. These places are sometimes listed, but where they are not, we are unable to determine whether that place is indeed Martigues, though it is the most likely. Where there was any uncertainty as to identities and names, this file served as a control.

The baptism data compiled by the genealogists are unfortunately not entirely exhaustive, some of them having yet to be analysed, even though the registers existed well before 1692 in three of the parishes and in L'Île between 1727 and 1735 (see S2 and S3 Figs).

As data are not currently available for all dates and the districts, we estimated the number of births by name and year for Martigues, combining all districts, both before and after 1720, as follows:

Let $q_j$ denote one of the four districts, $j \in [1..4]$.

We define two periods, $t_i$, $i \in [1..2]$, with P1, $t_1 = [1689–1720]$ and P2, $t_2 = [1721–1789]$

Let $d_{ji}$ represent the number of recorded years for district $j$ in period $i$; this number varies across districts.

Let $N_{sji}$ be the number of baptized individuals with a given name $s$, in district $q$, during period $t_i$.

The number $N_{si}$ of the baptized individuals with name $s$ in Martigues, $N_{si}$, for period $t_i$ (expressed per year) is estimated as:

$$N_{si} = \sum_{j=1}^{4} \frac{N_{sji}}{d_{ji}}$$

(1)

This involves estimating the number of the baptized individuals in years with missing data based on the counts from years for which data are available (see S4 File for calculation examples).

As we will show, there is a strong correlation between the total number of births per period and the index $N_{si}$, whether considering the name alone or including its spelling variations ($r > 0.97$). This high correlation reflects the relatively minor effect of lemmatization on the index.

## 2.2. The parish registers of Bouches-du-Rhône, provided by Geneanet

The baptism certificates provided by Geneanet also concern different municipalities or parishes in Bouches-du-Rhône from 1689 to 1890 ("GᴇɴB-BᴅR" file). These data can be used to locate the names in this department. They cover the periods P1 (1689–1720), P2 (1721–1789), P3 (1790–1840), and P4 (1841–1890). The data serve to verify whether the names present in Martigues, before and after the plague, are geohapax names (names present exclusively in a single geographical area relative to a set of geographical areas and for a given period) or names that can be found in the neighbouring municipalities of Martigues. The map in S5 Fig shows the distribution of births by municipality and year between 1689 and 1720. Birth numbers are highest in the municipalities of Aix-en-Provence, Marseilles and Martigues.

## 2.3. The INSEE surname file (1891–1915)

The surname data from the INSEE file (Institut National de la Statistique et des Études Économiques) ("D-ɪɴsᴇᴇ" file) [25] are also used to compare and locate names. By surname, they provide the number of births registered by municipalities between 1891 and 1915. These data cover much later periods than those in our study, but they follow on from the Martigues GᴇɴB-MAR corpus and the GᴇɴB-BᴅR corpus. Despite the diversity of sources, the comparisons remain relevant as interpretation aids in that the geographical specificity of the names and their lengthy existence have been largely demonstrated [8].

## 2.4. The file of plague deaths

Between October 1720 and June 1721 [20], historical sources point to 2,150 victims in Martigues [21], out of a population of 5,886 officially counted in 1716 (S1.2 in S1 File) The parish registers recorded the deaths in the

customary manner at the start of the epidemic and subsequently in the form of a list drawn up in the wake of the contagion. These documents provide the victims' surname, given name, sex, age, date and place of death, identity of the father and mother or spouse, occupation, and, in some cases, kinship information. It was not possible to retrospectively provide a name for all plague victims. For some, we know the surname and given name, while others remain entirely anonymous. Previous research [22,23] has provided a list of the plague victims in Martigues based on death certificates, for a total of 2,134 people. The list of plague victims' names is hereafter called D-M_DP files (see data file S7 File).

## 3. Lemmatization

The list of 4,022 surnames was drawn up before any attempt at lemmatization, which would consider minor spelling variations and errors in transcription in the databases, grouping all these variations of the same surname together in a single standard form. Lemmatization is particularly necessary here because the spelling of surnames is not set in stone and varies according to the person having written the names down, notably in parish registers. However, lemmatization remains a delicate process. Correctly homogenizing names with divergent spellings within the same family line calls for precise genealogical data over several generations and the control of the homogeneity of the spelling of the name of a single person where that name is reported in several registers. In the absence of data that would justify such lemmatizations, alternative strategies may be employed. These are discussed in detail in Supplementary Information S6 File. The final results are presented in the tables below.

## 4. Categories of names and results

To avoid the use of uncertain data and to facilitate analysis and discussion, we distinguish between four categories or types of names around which the analysis and discussion are organized: hapaxes (names represented by a single birth), Type 1 names present in P1 and absent in P2, Type 2 names present in P2 but absent in P1, and Type 3 names present both in P1 and P2. This categorization is difficult in some cases in that it may depend on the degree of lemmatization selected. This will be discussed each time it appears necessary.

### 4.1. Hapax

Hapaxes account for a particularly high proportion of the names, at 57.08% before 1721 and 50.30% after 1720. This high proportion, typical of all surname databases [1,2,26–28] results primarily from the fact that it is observed before lemmatization. It is difficult to determine whether the surnames of a single individual are true hapaxes, names given to a single baptized person, or what we could define as 'orthographic' hapaxes, i.e., names that are attributed to a single individual, but which already exist in another spelling variant (such as ABILLE, a hapax, but more likely "orthographic hapax" of ABEILLE). Since many feminized forms are to be found in just one or two examples, mainly before 1720, this may explain why hapaxes, including orthographic hapaxes, are rarer after 1721 than before. By aligning the feminine form with the 'masculine' form of the names, we could only reduce the number of hapaxes before 1721. It may also be the case the names were more rigorously transcribed towards the end of the study period than at the beginning, which would explain the higher number of hapaxes at the beginning of the period. It also needs to be stressed that hapaxes are unique names present in just one case over a long period of time: 32 years for the period before 1721 and 69 years for the period after 1720. It may be suggested that these single births over such a long period do not correspond, or correspond only slightly, to the names of family lines firmly established over time in Martigues, but at best are unique spelling variants or the name of a baptized person every now and then in Martigues. A simple interpretation of hapaxes thus remains highly unrealistic, which leads us to focus on other types of surnames that give a more reliable and informative image of the population of Martigues over the medium and long term.

## 4.2. Type 1

Type 1 names are the names given to people baptized between 1689 and 1720, but which later disappeared from Martigues. The list of such names given to over three births before 1721 is provided in Table 2. We selected this over-three based on the idea that the presence of at least four births over a 32-year period would enable the selection of names clearly established locally. To these names we have agglomerated their spelling variants (including feminized forms) and which are also those absent after 1720.

Thus, the form ARDISSON attributed to five births before 1721 and one afterwards is a Type 1 name, as well as the form ARDISON, with only one birth before 1721 (Table 2). Following the same reasoning, COUTURIÈRE, given to three births before 1721, and COUTURIER, given to one birth before 1721, were grouped under a single name, which explains the four COUTURIÈRE births in the table. In all such situations, described in Table 2, the variants of the names have thus been added to the most common form. Some names, such as COUPARD (two births, in 1692 and 1710), are Type 1 but are included among names with over three births only when adding the variants, such as COUPAR (two births, in 1699 and 1704) and COUPART (one birth in 1702).

We thus researched the presence of all Type 1 names in the file of plague deaths (D-Mdp). Some of these names (or their variants) are found in the file and marked 'p' in Table 2 (and in the following tables). Among the 93 Type 1 names, 50 are to be found in the file of plague deaths, or 53%.

Type 1 names identified in Martigues before 1721 are names that can also be found in Bouches-du-Rhône municipalities both before and after 1721. The name LANGLOIS, fairly common (15 births) before 1721 and absent after that date, are extremely common in Aix-en-Provence (with six births, all the children of a single couple), Aubagne (three LANGLOIS and one L'ANGLOIS) and Marseilles (one birth). After 1720, the name disappeared from Bouches-du-Rhône, except for three LANGLOIS born in Saint-Martin de Crau after 1841 (period P4). There is no doubt as to the geographical origin of these names given the distribution of LANGLOIS in France, concentrated at the end of the 19th century in Normandy. These births with a name suggesting English origins are present in Martigues at the end of the 17th century but later disappear. Were they wiped out by the plague, with just two children aged 4 years and 6 months included in the D-Mdp file? Did they leave the town? Or, more simply, did they survive but without having any descendants after 1720? These hypotheses are likely given the occupation of the fathers (three of them sailors, one a doctor, and one a 'bourgois').

As with the LANGLOIS, the 'ANGLE(s,z)(y(i)(e))', a name whose origin is in no doubt, were present in Martigues before the plague (24 births after lemmatization) and subsequently disappeared, while births with these names were particularly common, and almost exclusively occurring, in Auriol, throughout the period (71 births in P1 and 52 in P2). The occupation of the fathers of these ANGLEZY is given as 'worker' or rural labourer.

The four individuals baptized JAMBON in Martigues (three girls born in 1710, 1712, 1714 and one boy born in 1717) had the same mother (GABRIELLE Catherine) and father (Estienne, a sailor). Three of them died from the plague. This name is relatively rare and found outside Martigues in the P1 period, with one birth in Aix, one in Arles, one in Saint-Chamas, and two in Marseilles. After the plague, while the name disappeared from Martigues, it was to be found commonly in Arles (40), La Ciotat (18) and Saint-Chamas (6), before disappearing later. The occupation given for JAMBON in Martigues is sailor, while this occupation was not reported elsewhere (in particular in Saint-Chamas after 1748), or its mention was absent. It may be supposed that the death of the children and the possible abandonment of fishing led the JAMBON family to migrate from Martigues to Arles, La Ciotat or Saint-Chamas.

From the list of names in Table 2, it is possible to extract those present in Martigues in the P1 period (1689–1720) and which were not present in the municipalities in the same department in the same period (Table 3). They account for 32 of the 93 names in Table 2. They are geohapaxes. They are good markers, specific to Martigues, and provide unambiguous information on possible movements from Martigues to other destinations, from one period to the next.

Some Type 1 names and geohapaxes in Table 3 are not found in the P2 period (1721–1789) in Bouches-du-Rhône municipalities, and even further field. These are 'disappeared' family names. This is the case for DIEGUE/DIEGOU, common

Table 2. Type 1 surnames, with their various possible spelling variants. These surnames were given to over three baptized children between 1689 and 1720, and were then absent until 1789. N_1, N_1-Var and N_1-Tot are, respectively, the numbers of baptized children carrying the most common form of the name, those carrying variants of the name, and the sum of the two, N_1-Total/year, the latter sum being expressed by year, according to the formula 1. The p's indicate names reported as those of names of people having died from the plague in the file of plague deaths. In italics are controversial Type 1 names followed by couples of values relating to the birth headcounts in P1 (1689–1720) and P2 (1721–1789).

| Name_Type_1 | N_1 | N_1+var | Nb_1-Total | N_1/year | N_1-Tot/year | Plague | Variants | Name_Type_1 | N_1 | N_1+var | Nb_1-Total | N_1/year | N_1-Tot+var/year | Plague | Variants |
|---|---|---|---|---|---|---|---|---|---|---|---|---|---|---|---|
| ANGLESY | 16 | 8 | 24 | 0.5751 | 0.8568 | p | ANGLE(SE)(SI)(ZY) | HENRIQUE | 3 | 4 | 7 | 0.0938 | 0.2188 | n | HENRIGUE; HENRYGUE; *HENRIC:0,1* |
| DIEGUE | 14 | 8 | 22 | 0.4375 | 0.6875 | p | DIEGUOU | ROUILLON | 6 | 1 | 7 | 0.1875 | 0.2188 | n | ROUIOLLONE |
| LANGLOIS | 15 | 1 | 16 | 0.5697 | 0.6010 | p | LANGLOISE; *LANGLOIS: 5,1:birth date 1721* | ARDISSON | 5 | 1 | 6 | 0.1851 | 0.2163 | p | ARDISON |
| COUTET | 10 | 3 | 13 | 0.3688 | 0.4697 | p | COUTETTE | FARENQUE | 3 | 3 | 6 | 0.1154 | 0.2163 | p | FARENQ; FARENC; FAREN |
| JAUME | 9 | 4 | 13 | 0.3246 | 0.4640 | p | JAUMET; JAME | HERMITTE | 3 | 3 | 6 | 0.1053 | 0.2106 | p | HERMITE:1,0; HARMITE:2,0 |
| MAUREN | 11 | 2 | 13 | 0.3942 | 0.4639 | n | MAURENE(S) | MOURIES | 3 | 3 | 6 | 0.1053 | 0.2106 | n | MOURIER; MOURIERE |
| BREMON | 7 | 6 | 13 | 0.2476 | 0.4495 | p | BREMON(NE)(D); *BREMONT:0,1;* | BIRONNETTE | 3 | 2 | 5 | 0.1010 | 0.2091 | p | BIRONETTE; BIRONNET; *BIRONET:0,1* |
| JACQUES | 10 | 4 | 14 | 0.3197 | 0.4447 | p | JACQUE; JAQUE(S)(T) | CHAR-DOUSSE | 5 | 1 | 6 | 0.1635 | 0.2019 | n | CHARDOUSE |
| BEDOUIN | 10 | 2 | 12 | 0.3689 | 0.4429 | p | BEDOUINE | LAGARDE | 6 | 0 | 6 | 0.1947 | 0.1947 | p | |
| MATTY | 5 | 6 | 11 | 0.1923 | 0.4231 | p | MAT(H)Y(I) | LAUX | 4 | 2 | 6 | 0.1322 | 0.1947 | p | LAUS |
| BLANQUE | 9 | 3 | 12 | 0.3015 | 0.4168 | p | BLANQU(I)(IN) | AUBRET | 5 | 0 | 5 | 0.1923 | 0.1923 | n | |
| CLAVIER | 7 | 5 | 12 | 0.2332 | 0.4038 | n | CLAVIERE | COUPAR | 2 | 3 | 5 | 0.0769 | 0.1923 | p | COUPARD; COUPART |
| BARBIER | 10 | 1 | 11 | 0.3646 | 0.4016 | p | BARBIERE; *BARBIERI:0,1* | COUPARD | 2 | 3 | 5 | 0.0769 | 0.1923 | p | COUPAR; COUPART |
| GRAILLE | 8 | 1 | 9 | 0.3077 | 0.3462 | n | GRAILE | CHATAU | 4 | 1 | 5 | 0.1538 | 0.1923 | p | CHATEAU (*CHATAUD:p*) |
| REBATU | 8 | 1 | 9 | 0.3077 | 0.3462 | n | REBATTU | BROGLIA | 5 | 1 | 6 | 0.1190 | 0.1875 | p | BROGLA; *BROGLIE: p* |
| ESTAGNIER | 8 | 6 | 14 | 0.1538 | 0.3462 | p | ESTAGNIERE; ESTAGNER; *ESTAGNIER:0,3* | COURRAUD | 5 | 1 | 6 | 0.1563 | 0.1875 | p | COURRAUT; *COURRAND: p* |
| ROUSSE | 10 | 1 | 11 | 0.3125 | 0.3438 | p | ROUSE | COUGNET | 4 | 2 | 6 | 0.1250 | 0.1875 | n | COUIGNET; COUINETTE |
| ROUBIN | 8 | 0 | 8 | 0.3077 | 0.3077 | p | | GRATIAN | 3 | 3 | 6 | 0.0938 | 0.1875 | p | GRATIANE; GRATIANNE |
| ALIMAN | 7 | 1 | 8 | 0.2692 | 0.3077 | n | AALIMAN | RIMBAUD | 5 | 0 | 5 | 0.1866 | 0.1866 | p | |
| BENET | 7 | 1 | 8 | 0.2635 | 0.3020 | p | BENETE | ROUARD | 5 | 0 | 5 | 0.1852 | 0.1852 | n | |
| PONT | 5 | 3 | 8 | 0.1923 | 0.2861 | n | PON(TES); *PONT:0,1* | BRIVE | 5 | 0 | 5 | 0.1779 | 0.1779 | p | |
| AUDIFRET | 4 | 5 | 9 | 0.1250 | 0.2813 | p | AUDIF(F)RE(TT)E | GOUREL | 5 | 0 | 5 | 0.1765 | 0.1765 | n | |

*(Continued)*

Table 2. (Continued)

| Name_Type_1 | N_1 | N_1+var | Nb_1-Total | N_1/year | N_1-Tot/year | Plague | Variants |
|---|---|---|---|---|---|---|---|
| MICHAELIS | 3 | 6 | 9 | 0.0938 | 0.2813 | p | MICHAEL(I)(Y); MICHA(R)LIS; MICHAELLIER |
| RAYBAUD | 6 | 2 | 8 | 0.2091 | 0.2716 | n | RAYBAUDE |
| BOUTIER | 4 | 3 | 7 | 0.1538 | 0.2692 | p | BOUTIERE |
| IDA | 7 | 1 | 8 | 0.2260 | 0.2644 | p | IDADE |
| FARRAND | 4 | 3 | 7 | 0.1322 | 0.2644 | n | FARRAIN; FARRAN(DE) |
| COUNIL | 4 | 6 | 10 | 0.1250 | 0.2616 | n | COUNILLE; COUNINE |
| DUQUESNAY | 6 | 1 | 7 | 0.2222 | 0.2593 | n | DU QUESNAY |
| CORPORAL | 6 | 3 | 9 | 0.1875 | 0.2500 | n | CORPORA(LLE)(NNE) |
| TENOUX | 4 | 4 | 8 | 0.1250 | 0.2500 | n | ATENOUX; THENOUX |
| TOUFANY | 3 | 5 | 8 | 0.0938 | 0.2500 | p | TOUPHANY; TOUPHANI |
| VAILLEN | 5 | 3 | 8 | 0.1563 | 0.2500 | n | VAILLENQUE; VAILLAN(T)(E)(QUE) |
| ISOARD | 3 | 4 | 7 | 0.1154 | 0.2476 | p | ISOIRD; ISOUARD(E) |
| MALAVART | 1 | 6 | 7 | 0.0385 | 0.2476 | p | MALAUAR(D)(T); MALAVA(L)(E)(RDE) |
| MAYEN | 6 | 2 | 8 | 0.2019 | 0.2332 | p | MAYENE |
| ROUSSON | 4 | 3 | 7 | 0.1394 | 0.2332 | p | ROUSSONNE |
| BARGIER | 5 | 2 | 7 | 0.1692 | 0.2317 | p | BARGIERE; BARGI:0,1 |
| PERIAT | 2 | 4 | 6 | 0.0769 | 0.2308 | n | PERRIA(L)(S)(T) |
| NOUVEN | 6 | 0 | 6 | 0.2308 | 0.2308 | n | NOUVEU:0,2 |
| GRIFFE | 5 | 1 | 6 | 0.1923 | 0.2308 | n | GRIFE |
| MATERON | 5 | 1 | 6 | 0.1923 | 0.2308 | n | MATERONNE; MATHERON:0,2 |
| COUTELIER | 3 | 3 | 6 | 0.1154 | 0.2308 | p | COUTELIERE |
| BOUNEL | 3 | 3 | 6 | 0.1154 | 0.2236 | n | BOUNELLE |
| ROCHEFORT | 6 | 0 | 6 | 0.2222 | 0.2222 | n | |
| DEFONTE | 3 | 3 | 6 | 0.1082 | 0.2221 | n | DEFON; DEFONT; DEFONS |
| PIGNON | 5 | 2 | 7 | 0.1563 | 0.2188 | n | PIGNONNE |

| Name_Type_1 | N_1 | N_1+var | Nb_1-Total | N_1/year | N_1-Tot+var/year | Plague | Variants |
|---|---|---|---|---|---|---|---|
| TASSELIN | 5 | 0 | 5 | 0.1736 | 0.1736 | p | TASSEL:0,8 |
| MAILLEAU | 2 | 3 | 5 | 0.0625 | 0.1736 | p | MAILLEFAU(D); MAILLEFORT |
| ARMIEU | 4 | 1 | 5 | 0.1322 | 0.1707 | n | ARMIOU |
| MOURGUES | 5 | 0 | 5 | 0.1635 | 0.1635 | p | MOURGUE: 4,1; MOURGUI(E)N:0,2 |
| DESEGAU | 5 | 0 | 5 | 0.1563 | 0.1563 | n | |
| JURE | 5 | 0 | 5 | 0.1563 | 0.1563 | n | |
| VACQUE | 4 | 1 | 5 | 0.1250 | 0.1563 | p | VAQUE: p |
| DEBECARIS | 3 | 2 | 5 | 0.0938 | 0.1563 | n | DEBECARY; DEBEUCARIS |
| BONNETON | 4 | 0 | 4 | 0.1538 | 0.1538 | p | |
| CASTREUIL | 4 | 0 | 4 | 0.1538 | 0.1538 | n | |
| JAMBON | 4 | 0 | 4 | 0.1538 | 0.1538 | p | |
| SAUNIER | 3 | 1 | 4 | 0.1154 | 0.1538 | p | SAUNIERE |
| NATE | 2 | 2 | 4 | 0.0769 | 0.1538 | n | NATTE |
| SPILALIER | 1 | 3 | 4 | 0.0385 | 0.1538 | n | SPITALIER€; SPITALLIER |
| DELARMET | 4 | 0 | 4 | 0.1524 | 0.1524 | n | |
| MORE | 4 | 0 | 4 | 0.1481 | 0.1481 | n | |
| ROUNIER | 3 | 1 | 4 | 0.1111 | 0.1481 | n | ROUNIERE |
| ROQUET | 3 | 1 | 4 | 0.1125 | 0.1438 | p | ROQUE |
| BOET | 4 | 0 | 4 | 0.1322 | 0.1322 | n | |
| MINARD | 4 | 0 | 4 | 0.1322 | 0.1322 | p | |
| BUS | 4 | 0 | 4 | 0.1308 | 0.1308 | n | |
| MERCURIN | 2 | 2 | 4 | 0.0683 | 0.1308 | n | MERCURIN |
| CHAMBARD | 3 | 1 | 4 | 0.0938 | 0.1250 | n | CHAMBARDE |

**Table 3. List of Type 1 geohapax surnames in Martigues.** They are extracted from Table 2, with the number of baptized individuals in P1 (1689–1720) and P2 (1721–1789). The municipalities (INSEE code) in which the names of the baptized are found in P2 and Bouches-du-Rhône, with their number between brackets. In italics are the names with uncertain classifications, followed by the number of births in P1 and P2. The $p$ value holds for the names and their variants listed in the file of plague deaths (D-M$_{DP}$).

| Geohapax Type_1 name | 1689-1720 | 1721-1789 | Plague | |
|---|---|---|---|---|
| DIEGUE, DIEGOU | 22 | 0 | p | |
| REBATU, REBATTU | 9 | 4 | n | 13070-La Penne-sur-Huveaune (3); 13028-La Ciotat (1) |
| ALIMAN, AALIMAN | 8 | 3 | n | 13051-Lançon-de-Provence (3) |
| CLAVIER, CLAVIERE | 12 | 1 | n | 13113-Venelles (1) |
| IDA, IDADE | 8 | 0 | p | |
| CORPORA(L)(LLE)NNE) | 8 | 0 | n | |
| DUQUESNAY, DU QUESNAY | 7 | 0 | n | |
| ROUILLON, ROUILLONNE | 7 | 0 | n | |
| AUBRET | 5 | 0 | n | |
| BRIVE | 5 | 0 | p | |
| BROGLIA, BROGLA | 6 | 0 | p | |
| CHARDOUSSE | 6 | 9 | n | 13103-Salon-de-Provence (9) |
| COURRAUD, COURAUD | 6 | 4 | p | 13082-Rognes (3); 13028-La Ciotat (1) |
| DESEGAU | 5 | 0 | n | |
| GRIFFE, GRIFE | 5 | 23 | n | 13004-Arles-Saint-Julien (1); 13058-Maussane-les-Alpilles (19); 13028-La Ciotat (3) |
| JURE | 5 | 0 | n | |
| BONNETON | 4 | 4 | p | 13114-Ventabren (1); 13032-Eguilles (3) |
| CASTREUIL | 4 | 0 | n | |
| CHATAU, CHÂTEAU | 5 | 4 | p | 13005-Aubagne (3); 13007-Auriol (1) |
| ESTAGNIERE, ESTAGNER, *ESTAIGNIER:0,3* | 9 | 3 | p | |
| LAUX, LAUS | 6 | 0 | p | |
| MINARD | 4 | 0 | p | |
| VACQUE | 5 | 0 | p | |
| BIRONNETTE, BIRONNET, *BIRONET:0,1* | 5 | 1 | p | |
| COUTELIERE, COUTELIER | 6 | 0 | p | |
| DEBECARIS, DEBECARY, DEBEUCARIS | 5 | 0 | n | |
| DEFONTE, DEFON, DEFONT, DEFONS | 6 | 1 | n | 13039-Fos-sur-Mer (1) |
| GRATIAN, GRATIANE, GRATIANNE | 6 | 0 | p | |
| ROUNIER, ROUNIERE | 4 | 0 | n | |
| TOUFANY, TOUPHANY, TOUPHANI | 8 | 3 | p | 13004-Arles - Saint-Julien (3) |
| COUPAR, COUPARD, COUPART | 5 | 0 | p | |

enough in P1, with 22 baptisms, but missing entirely in P2. The fathers of the DIEGUE and DIEGOU births were all sailors or fishing masters, which may explain their mobility after the epidemic, especially since the surnames of their mothers are not typical of the region RIVIERE, RENOUE, etc.). At least four DIEGUE are included in the list of plague deaths. Neither are these names to be found in the INSEE file between 1891 and 1915 (though there were three births in the Pyrénées-Atlantiques department between 1916 and 1940, probably from a completely different origin).

The ROUILLON surname also disappeared in P2 from Bouches-du-Rhône and is not found in the D-MDP file, even though it is a fairly common name in France, in departments as diverse as Eure-et-Loir and Vosges. The BROGLIA surname is listed in the D-Mdp file but not in P2 in Bouches-du-Rhône, though it is a common surname in south-east France (D-INSEE file). The AUBRET surname is found mainly in Vendée and Indre, as well as in Loire and Saône-et-Loire.

GRATIAN(NE) is also a Martigues surname. Six births are recorded in the Gen-Bmar file (1697, 1700, 1702, 1704, 1708, 1711), the children of two brothers identified as "doctors of medicine". None of these six children appear to have died from the plague; only the father, aged 70, is listed in the file of plague deaths. We can thus conclude that the GRATIAN disappeared after 1720 from Martigues, and, according to our data, from Bouches-du-Rhône, not because they died from the plague but because they fled elsewhere. According to INSEE data for 1891–1915, the GRATIAN were located primarily in south-west France (Gironde, Gers, Pyrénées-Atlantiques), with just three births in Marseilles.

Another category of geohapax in Martigues in P1 (1689–1720) can be found in other Bouches-du-Rhône municipalities in P2, though they were not recorded there in P1. One such surname in this category is CHARDOUSSE, common in P1 in Martigues, with six births, and found in P2 in Salon-de-Provence (nine births). This surname is also found (D-insee) in the Var and Alpes-de-Haute-Provence departments. Other geohapax surnames in this category are BONNETON, missing from Martigues after 1721 but present in Ventabren and Eguilles after 1721, and CHATAU, found in P2 in Aubagne and Auriol.

Table 3 shows that 50% of its constituent names are also to be found in the register of plague deaths (D-Mdp), slightly less than the total of Type 1 names in Table 2. Thirty per cent of these geohapaxes can be found in another Bouches-du-Rhône municipality.

### 4.3. Type 2

Type 2 names are names unknown in Martigues before 1721 (1689–1720) but found there after 1720 (1721–1789). They correspond to the names of newly baptized individuals arriving in Martigues after the plague epidemic. This category helps to infer the geographical origin of people settling in Martigues after the epidemic by determining their presence in the surrounding municipalities of Bouches-du-Rhône and across France. After lemmatization, these names are 154; only the first 100 of these names, which all include at least four baptized individuals, are provided in Table 4.

The arrival in Martigues of these surnames given to new births reflects an emphatic renewal of the population after 1720, even if our conclusions remain dependent on the degree of lemmatization, which could shift surnames from the Type 2 category to the Type 3 category.

Some of these Type 2 surnames are identified elsewhere in the municipalities of the Bouches-du-Rhône department, while others clearly show a more distant origin. Each name thus appears to have a particular geographical origin. A few examples will be discussed here.

Some surnames could not be included in the Type 2 category because they are represented in P1 only by one or two births, while the latter are plentiful in P2. Strictly speaking, they should be included in the Type 3 category (see below). This is the case with GERMAIN, for which two baptisms are identified in the first period (GERMAN in 1689 and 1696), rising to 15 in the second period (all GERMAIN). A further such case is SEREN/SEREM/SEREIM. This name is extremely common in P2 (24 births), but there was just one SEREIM birth in P1, in 1689.

The classification of the PIGNATEL surname is equally debatable. Under this spelling, it is effectively a Type 2 name, but once lemmatized with the numerous variants, it would be classified more as a Type 3 name. There were three PINATEL(LE)

**Table 4. List of 100 Type 2 surnames.** They were absent in Martigues between 1689 and 1720 (P1) but present in P2 (1721–1789) with the number N_2 of baptisms for each name, the number of their variants, N_2- Var, and for the sum of the two, N_2-Total, with the total number of baptized calculated per year for the period, P2 N_2-Total/year. In italics are controversial Type 2 names followed by couples of names relating to the birth headcounts in P1 and P2.

| NAME | N_2 | N_2-Var | N_2-total | N_2/year | N_2-total/year | Variants | NAME | s | N_2-Var | N_2-total | N_2/year | N_2-total/year | Variants |
|---|---|---|---|---|---|---|---|---|---|---|---|---|---|
| PIGNATEL | 85 | 21 | 106 | 1.468 | 1.842 | PIGNAT(ELLE)(ELLY); *PINAT(EL)(ELLE):3,30* | GUICHARD | 11 | | 11 | 0.164 | 0.164 | |
| MAURAS | 12 | 20 | 32 | 0.226 | 0.604 | *MAURRAS:1 (1701),20* | QUEIREL | 4 | 7 | 11 | 0.058 | 0.164 | QUIEREL; QUIREL; QUIREOU; QUIREU |
| CAMBON | 29 | | 29 | 0.477 | 0.477 | | MALAGET | 9 | 2 | 11 | 0.130 | 0.159 | MALAJET; MALAGER |
| LATAUD | 25 | 2 | 27 | 0.428 | 0.461 | LATAU;LATOUD | GARROT | 11 | | 11 | 0.159 | 0.159 | |
| POUCEL | 25 | 7 | 32 | 0.362 | 0.464 | PONCEL; POUCEAU; POUCEOU | LIEURON | 11 | | 11 | 0.159 | 0.159 | |
| CHALVE | 26 | 4 | 30 | 0.377 | 0.435 | CHALVI | PARET | 7 | 2 | 9 | 0.123 | 0.153 | PARES |
| SEREN | 22 | 2 | 24 | 0.319 | 0.349 | SEREIN; *SEREM:1,0* | CABUS | 8 | | 8 | 0.151 | 0.151 | |
| SOUREIL-LET | 21 | 1 | 22 | 0.304 | 0.319 | SOURREILLET | DALMAS | 8 | | 8 | 0.151 | 0.151 | |
| MONGIN | 18 | 3 | 21 | 0.266 | 0.309 | MONGE; MONGES | VILLAMUS | 7 | 1 | 8 | 0.132 | 0.151 | VILLANUS |
| GUES | 15 | 1 | 16 | 0.283 | 0.302 | GUEX | REIMON-DON | 4 | 6 | 10 | 0.060 | 0.149 | REIMONDON(E)(NE) |
| FOUQUES | 20 | | 20 | 0.299 | 0.299 | *FOUQUE: type_3* | FENON | 6 | 4 | 10 | 0.087 | 0.145 | FENOU |
| MOURRE | 16 | 3 | 19 | 0.245 | 0.293 | MOURE; MOURRES | TARDY | 7 | 2 | 9 | 0.115 | 0.144 | TARDI |
| GUIDON | 16 | 1 | 17 | 0.258 | 0.273 | GUIDOUN | FLAMEN | 7 | | 7 | 0.132 | 0.132 | FLAME:1,0 |
| ANTEOUME | 14 | 4 | 18 | 0.203 | 0.270 | ANTEAUME | MOURET | 7 | | 7 | 0.132 | 0.132 | |
| TOURNET | 9 | 6 | 15 | 0.152 | 0.266 | TOURNEL | PARPAN | 7 | | 7 | 0.132 | 0.132 | |
| BOUNIN | 18 | | 18 | 0.261 | 0.261 | | SAMUEL | 9 | | 9 | 0.130 | 0.130 | |
| AUZIERE | 15 | 2 | 17 | 0.217 | 0.246 | AUZIER | GOUR-GUES | 8 | 1 | 9 | 0.116 | 0.130 | GOURGUE |
| MAURAN | 9 | 5 | 14 | 0.152 | 0.242 | MAURAND; MAURANS | ARMAG-NIER | 5 | 2 | 7 | 0.094 | 0.128 | ARMAINIER; ARMAGNE |
| SUBE | 8 | 7 | 15 | 0.125 | 0.227 | SUBI;SUBY; *SUBLI:1,0* | LONG | 8 | | 8 | 0.125 | 0.125 | |
| CARLON | 12 | | 12 | 0.226 | 0.226 | | AUVET | 8 | | 8 | 0.116 | 0.116 | |
| MERON | 12 | | 12 | 0.226 | 0.226 | | CODE | 8 | | 8 | 0.116 | 0.116 | |
| TURIN | 12 | | 12 | 0.226 | 0.226 | | NALIN | 8 | | 8 | 0.116 | 0.116 | |
| GOIRAN | 12 | 3 | 15 | 0.178 | 0.226 | GOIRAND:1(1697),3 | VADON | 8 | | 8 | 0.116 | 0.116 | |
| BARBOT | 15 | | 15 | 0.217 | 0.217 | | GRIGNAN | 7 | 1 | 8 | 0.101 | 0.116 | GRIGNON |
| GERMAIN | 15 | | 15 | 0.217 | 0.212 | GERMAN:1,0; GERMANE:1,0 | PETRACHE | 7 | 1 | 8 | 0.101 | 0.116 | PETRACHES |
| TERLIER | 14 | | 14 | 0.212 | 0.212 | | PEIRON-CELY | 6 | 2 | 8 | 0.087 | 0.116 | PE(Y)RONCELY |
| BAUZAN | 11 | | 11 | 0.208 | 0.208 | | RODEZ | 6 | 2 | 8 | 0.087 | 0.116 | RODES; RODHES |
| BRES | 11 | | 11 | 0.208 | 0.208 | | ANDRIEU | 7 | | 7 | 0.115 | 0.115 | |
| CHASSEL-OUP | 11 | | 11 | 0.208 | 0.208 | *CHASSELOU:1,1; CHASSELOUN(Z)E:2,0* | ARTAUD | 5 | 2 | 7 | 0.077 | 0.115 | ARTAU |
| CASSOLE | 10 | 1 | 11 | 0.189 | 0.208 | CASSOLA | PREVOT | 6 | | 6 | 0.113 | 0.113 | |
| AVON | 14 | | 14 | 0.207 | 0.207 | | SOUQUET | 6 | | 6 | 0.113 | 0.113 | |
| PIPIN | 14 | | 14 | 0.203 | 0.203 | | VIEL | 6 | | 6 | 0.113 | 0.113 | |
| SURLIER | 5 | 9 | 14 | 0.072 | 0.203 | SURLIE; SURLIET; SURILLIER; SURLY | ESTELLE | 6 | | 6 | 0.109 | 0.109 | |
| DELESTI | 7 | 7 | 14 | 0.101 | 0.203 | DELESTI(C)(Y)(C) | PURPAN | 6 | | 6 | 0.109 | 0.109 | |
| FERRAIRE | 12 | | 12 | 0.196 | 0.196 | | LAMBERT | 7 | | 7 | 0.106 | 0.106 | |
| FREGIER | 11 | | 11 | 0.190 | 0.190 | | BREIGNE | 5 | 1 | 6 | 0.086 | 0.104 | BREIGUE |

*(Continued)*

**Table 4.** (Continued)

| NAME | N_2 | N_2-Var | N_2-total | N_2/year | N_2-total/year | Variants | NAME | s | N_2-Var | N_2-total | N_2/year | N_2-total/year | Variants |
|------|-----|---------|-----------|----------|----------------|----------|------|---|---------|-----------|----------|----------------|----------|
| AUDOYER | 10 | | 10 | 0.189 | 0.189 | | GASSIER | 7 | | 7 | 0.101 | 0.101 | |
| BOULFARD | 8 | 5 | 13 | 0.116 | 0.188 | BOUL(L)(E)FAR(D)T; BOULPHAR(D) | SANTON | 7 | | 7 | 0.101 | 0.101 | |
| BARGE | 7 | 5 | 12 | 0.101 | 0.183 | BARJE;BARGES | VACHEN | 7 | | 7 | 0.101 | 0.101 | |
| REGIMBAUD | 6 | 6 | 12 | 0.087 | 0.178 | REGIBAUD; REGINAUD | AUGIER | 6 | | 6 | 0.096 | 0.096 | |
| AMPHOUX | 10 | 2 | 12 | 0.145 | 0.174 | AMPHOUS; AMPHOUXNE | BONNEFOY | 6 | | 6 | 0.096 | 0.096 | |
| GALICIAN | 8 | 4 | 12 | 0.116 | 0.174 | GAL(L)I(CIEN)(SSI(C)AN | BRESSIER | 5 | | 5 | 0.094 | 0.094 | |
| MICHELON | 11 | 1 | 12 | 0.159 | 0.174 | MICHELLON | BRIAND | 5 | | 5 | 0.094 | 0.094 | |
| CAMUS | 11 | | 11 | 0.173 | 0.173 | | DEFORT | 5 | | 5 | 0.094 | 0.094 | |
| GUIOU | 10 | 1 | 11 | 0.154 | 0.173 | GUILLOU | JERLIER | 5 | | 5 | 0.094 | 0.094 | |
| GUIBAUD | 9 | | 9 | 0.170 | 0.170 | | MORAND | 5 | | 5 | 0.094 | 0.094 | |
| CAIRE | 8 | 1 | 9 | 0.151 | 0.170 | CAIRET | PETRE | 5 | | 5 | 0.094 | 0.094 | |
| CHAUTARD | 8 | 1 | 9 | 0.151 | 0.170 | CHAUTART | BOSSY | 4 | 1 | 5 | 0.075 | 0.094 | BOSSI |
| RIGOND | 7 | 3 | 10 | 0.123 | 0.167 | RIGOID; RIGONE | DANGLA | 4 | 1 | 5 | 0.075 | 0.094 | DANGLADE |
| DAVIN | 10 | | 10 | 0.164 | 0.164 | | MIRONET | 3 | 2 | 5 | 0.057 | 0.094 | MIRONNET; MIROUNET |
| CHABANAT | 11 | | 11 | 0.164 | 0.164 | | CHABAS | 5 | | 5 | 0.077 | 0.077 | |

births in P1 but 85 in P2, and there were also numerous PINATEL before 1689. This is why PIGNATEL is included both in Table 4 and in Table 5 of Type 4 surnames. This surname will be discussed later.

The CAMBON surname, absent in P1, was common in Martigues in P2, with 29 births (Table 4). The name is almost entirely absent in the Bouches-du-Rhône department before 1721, except for two baptisms in Marseilles in P1. This unique presence of CAMBON in Martigues is verified at least until 1800, after which it was to be found only in Marseilles (11 births between 1891 and 1915 and 16 births between 1916 and 1940) (Fig 2). The departmental distribution of CAMBON is concentrated in the Occitanie region (Hérault and Gard departments, see map). This distribution suggests that people with the CAMBON surname arrived after the plague, though they failed to take root, as the surname then disappeared from Martigues, and from the department, after 1780. The fathers of CAMBON births in Martigues were mainly bakers or oven workers, with two sailors recorded. It would be tempting to consider CAMBON and JAMBON as one and the same surname, but the distribution of the two names is different. In addition, the presence of a couple formed of CAMBON Charles and JAMBON Rose, who had ten CAMBON births in the Ferrières district of Martigues between 1730 and 1750, suggests that the lemmatization of CAMBON and JAMBON created confusion between the two distinct lines, although this could perhaps be the result of an intermarriage?

The LATAUD surname is present almost exclusively in Grans ($N = 10$ and $N = 47$, before and after 1720) (Fig 3). The LATAUD families were farmers, growers, carders and bakers. Numerous LATAUD arrived in Martigues after 1720 ($N = 27$), principally becoming sailors or pulley-men. But their presence in Martigues was short-lived, as none are to be found there after 1789. The distribution of the surname was highly concentrated in Bouches-du-Rhône; according to the D-INSEE file, the department was home to 76% of LATAUD births between 1891 and 1915.

The ANTEAUME surname is classified as Type 2, with some reservations. It can be found in Martigues after 1720, along with ANTEOUME in the Ferrières district. The name was present in Arles before 1720. Should this name be considered as a variant of ANTELME/ANSELME (a single birth before 1720 and ten later, mostly in the Jonquières district), as a variation of the same names with a 'g' prefix GANTAUME, GANTEAUME, GANTELME; 16 births before 1720, four after), or as a variation of

**Table 5. Distribution of 60 Type 3 surnames. The first 30 showed the strongest annual increase in the number of births before the plague (P1: 1689–1720) and after the plague (P2: 1721–1789), followed by the distribution of the 30 surnames with the strongest annual decrease in the number of births before (P1) and after (P2) the plague of 1720. N_1 and N_2 are the number of births per name listed in the first column, and N_1+var and N_2+var include variants. These same last numbers are expressed per year (N_1+var/year and N_2+var/year). The two following columns are the differences and ratio between these two numbers in P2 and P1. The p's indicates a name included in the file of plague deaths (D-MDP). The cases of TOURRE/TOURREL and PIGNATEL are debatable. The BRASSAVIN surname ranks 36th in terms of the strongest increase in births between the two periods.**

| NAME | N_1 | N_2 | N_1+Nvar | N_2+Nvar | Ni_1+Nvar/year | Ni_2+Nvar/year | (N_2+Nvar/year)-(N_1+Nvar/year) | (N_2)+Nvar/year)/(N_1+Nvar/year) | Plague | Variants |
|---|---|---|---|---|---|---|---|---|---|---|
| TOURREL | 31 | 253 | 52 | 272 | 1.950 | 4.152 | 2.203 | 2.130 | p | TOUR(R)EL(L)E; TOUREL |
| FOUQUE | 223 | 630 | 223 | 650 | 7.659 | 9.638 | 1.979 | 1.258 | p | FOUQUES |
| PIGNATEL | 0 | 85 | 3 | 106 | 0.107 | 1.842 | 1.736 | 17.260 | n | PIGNAT(ELLE)(ELLY); PINAT(EL)(ELLE) |
| AUDIBERT | 61 | 257 | 71 | 263 | 2.499 | 4.032 | 1.533 | 1.614 | p | AUDIBER(D); AUDIBERTE |
| LAUGIER | 17 | 131 | 19 | 131 | 0.670 | 2.069 | 1.399 | 3.086 | p | LAUGIERE |
| CAMOIN | 13 | 102 | 20 | 130 | 0.625 | 1.944 | 1.319 | 3.110 | n | CAMOUIN |
| ROUBIEU | 11 | 57 | 21 | 141 | 0.808 | 2.052 | 1.245 | 2.541 | p | ROUBI(EU)OU; ROUBI(ER)(EUINE)(EUX)(ON)(OU) |
| GIDE | 45 | 186 | 49 | 191 | 1.623 | 2.824 | 1.202 | 1.741 | p | GIDDE; GIDES |
| PHALIPON | 1 | 71 | 3 | 87 | 0.115 | 1.280 | 1.164 | 11.090 | n | PHALIIP)(PS); PHALY(S); PHALIP(ON)(ONNE)(OUN)(PON); PHARIPON |
| ROUBIOU | 1 | 73 | 2 | 75 | 0.077 | 1.087 | 1.010 | 14.130 | p | ROUBION |
| PONCHIN | 20 | 121 | 21 | 122 | 0.808 | 1.800 | 0.992 | 2.229 | p | PONCHINE; PONCHON |
| ESPANET | 4 | 80 | 5 | 81 | 0.192 | 1.178 | 0.986 | 6.127 | p | ESPANNETE |
| PONCHIN | 20 | 121 | 21 | 121 | 0.808 | 1.786 | 0.978 | 2.211 | p | PONCHINE |
| CHOUQUET | 6 | 79 | 8 | 79 | 0.264 | 1.179 | 0.915 | 4.459 | n | CHOUQUETTE |
| CATELIN | 9 | 67 | 10 | 87 | 0.385 | 1.270 | 0.885 | 3.301 | n | CATELAN; CAT(H)EL(L)IN |
| HUGUES | 3 | 55 | 3 | 56 | 0.111 | 0.893 | 0.782 | 8.040 | p | HUGUE |
| HUGUES | 3 | 55 | 0 | 0 | 0.111 | 0.893 | 0.782 | 8.040 | p |  |
| BERAUD | 8 | 66 | 9 | 67 | 0.306 | 1.026 | 0.720 | 3.354 | p | BERAUDE |
| GONFARD | 1 | 6 | 3 | 50 | 0.094 | 0.744 | 0.650 | 7.937 | n | GONFAR; GONFARDE |
| ARNOUX | 11 | 66 | 11 | 66 | 0.390 | 1.027 | 0.637 | 2.633 | p | ARNOUX |
| VERAN | 9 | 46 | 10 | 63 | 0.313 | 0.939 | 0.627 | 3.006 | n | VERAND€; VERAN(NE) |
| GALLON | 1 | 25 | 11 | 62 | 0.377 | 0.990 | 0.613 | 2.627 | p | GALLONE; GALON(E)(NE) |
| GALON | 8 | 36 | 11 | 62 | 0.377 | 0.990 | 0.613 | 2.627 | p | GALLON€(NE); GALONE |
| AILLAUD | 8 | 58 | 14 | 66 | 0.435 | 1.048 | 0.613 | 2.410 | p | AIL(H)(L)AUD(E)(EAUD) |
| MISTRAL | 11 | 62 | 11 | 64 | 0.365 | 0.973 | 0.608 | 2.664 | p | MISTRALE |
| MISTRAL | 11 | 62 | 11 | 63 | 0.365 | 0.955 | 0.589 | 2.613 | p | MISTRALE |
| ESCAVI | 5 | 55 | 7 | 58 | 0.259 | 0.845 | 0.586 | 3.259 | n | ESCAV(I)(Y)(T) |
| FALIPON | 15 | 5 | 22 | 96 | 0.839 | 1.407 | 0.568 | 1.677 | p | (F)(PH)ALIP(PE); (F)(PH)ALIPOUN; (F)(PH)ALIPPON |
| MAURRAS | 1 | 20 | 1 | 32 | 0.038 | 0.604 | 0.565 | 15.698 | n | MAURAS |
| PARANQUE | 1 | 30 | 1 | 37 | 0.031 | 0.571 | 0.540 | 18.279 | p | PARENQUE; PARANGUE(Y) |
| ... | ... | ... | ... | ... | ... | ... | ... | ... | ... | ... |
| VACHIER | 37 | 21 | 41 | 27 | 1.339 | 0.398 | −0.941 | 0.297 | p | VACHIERE; VASCHIER(E)(ES) |
| MARTIN | 104 | 207 | 118 | 209 | 4.213 | 3.270 | −0.944 | 0.776 | p | MARTEN; MARTIN(N)E |

*(Continued)*

| NAME | N_1 | N_2 | N_1+Nvar | N_2+Nvar | Ni_1+Nvar/year | Ni_2+Nvar/year | (N_2+Nvar/year)-(N_1+Nvar/year) | (N_2+Nvar/year)/(N_1+Nvar/year) | Plague | Variants |
|---|---|---|---|---|---|---|---|---|---|---|
| VENCE | 22 | 18 | 33 | 19 | 1.268 | 0.280 | −0.988 | 0.221 | p | VENSE |
| PAIGNON | 14 | 2 | 29 | 2 | 1.029 | 0.033 | −0.996 | 0.032 | p | PAGNON(E)(NE); PAIGNON(E)(NE) |
| RICARD | 37 | 37 | 50 | 37 | 1.656 | 0.637 | −1.020 | 0.384 | p | RICARDE |
| PONS | 46 | 46 | 46 | 46 | 1.753 | 0.702 | −1.052 | 0.400 | p | |
| GOUIRAN | 51 | 25 | 68 | 80 | 2.380 | 1.321 | −1.059 | 0.555 | p | GOUIRAN(CE)(E)(NE)(T); GO(U)YRAN(D) |
| BRILLAND | 22 | 19 | 47 | 28 | 1.526 | 0.467 | −1.059 | 0.306 | p | BRILAN(D); BRILL(L)AN(DE)(T); BRILLOND |
| DURAND | 49 | 55 | 65 | 87 | 2.372 | 1.296 | −1.076 | 0.546 | p | DURAN; DURANDE; DURANT; DURANTE |
| SAUVAIRE | 23 | 22 | 27 | 22 | 1.445 | 0.363 | −1.082 | 0.251 | p | SAUVAIRIS; SAUVERIS |
| NUIRATTE | 27 | 4 | 41 | 14 | 1.317 | 0.220 | −1.097 | 0.167 | p | NUIRATE; NUISATTE; NURA(T)TE; NURIATTE |
| GRANIER | 66 | 84 | 73 | 84 | 2.495 | 1.392 | −1.102 | 0.558 | p | GRANIERE |
| BARTHELEMY | 42 | 29 | 47 | 37 | 1.691 | 0.580 | −1.111 | 0.343 | p | BARTELEMY; BARTH(ALAIS)(ELAIS)(ELEMI)(ELERM) |
| BELLON | 29 | 39 | 64 | 67 | 2.180 | 1.054 | −1.127 | 0.483 | p | BELLON(NE)(Y); BELON(E)(I)(NE)(Y) |
| COULET | 64 | 64 | 64 | 64 | 2.250 | 1.111 | −1.139 | 0.494 | p | |
| BLAY | 5 | 6 | 13 | 6 | 1.370 | 0.174 | −1.196 | 0.127 | n | BLAUY; BLAVY; BLAYDE; BLAYET; BLAYS |
| ROMAN | 52 | 48 | 54 | 51 | 2.028 | 0.827 | −1.201 | 0.408 | p | ROMAND; ROMAN(E)(S) |
| ABEILLE | 48 | 29 | 49 | 29 | 1.659 | 0.451 | −1.208 | 0.272 | p | ABILLE |
| TENQUE | 37 | 3 | 37 | 3 | 1.264 | 0.048 | −1.217 | 0.038 | p | |
| LIEUTAUD | 40 | 33 | 48 | 34 | 1.794 | 0.510 | −1.284 | 0.284 | p | LIEUTARD; LIEUTAUDE |
| BOUYER | 30 | 2 | 39 | 3 | 1.412 | 0.048 | −1.364 | 0.034 | p | BOUIER; BOUIERE; BOUYERE |
| GAUTIER | 86 | 138 | 108 | 140 | 3.780 | 2.235 | −1.545 | 0.591 | p | GAUTIERE; GAUTHIER |
| CAUDIERE | 53 | 27 | 54 | 28 | 1.972 | 0.423 | −1.549 | 0.215 | p | CAUDIER; CAUDRERE |
| BERTRAND | 57 | 55 | 75 | 55 | 2.508 | 0.863 | −1.646 | 0.344 | p | BERTRAN(E)(NE); BERTRANDE |
| ANTOINE | 77 | 56 | 86 | 94 | 3.221 | 1.432 | −1.789 | 0.445 | p | ANTHOINE |
| ROUSSIN | 59 | 29 | 75 | 29 | 2.616 | 0.429 | −2.187 | 0.164 | p | ROUSSINE |
| BEAUMON | 29 | 9 | 85 | 54 | 3.073 | 0.813 | −2.260 | 0.265 | p | BEAUMON(D)(E)(NE)(T) |
| VENEL | 88 | 78 | 108 | 78 | 4.062 | 1.165 | −2.896 | 0.287 | p | VENEL(L)E |
| TOURRE | 111 | 69 | 111 | 69 | 4.038 | 1.008 | −3.030 | 0.250 | p | |
| ARNAUD | 103 | 96 | 125 | 96 | 4.523 | 1.457 | −3.066 | 0.322 | p | ARNAUDE |

# CAMBON

[1689-1720]

[1721-1789]

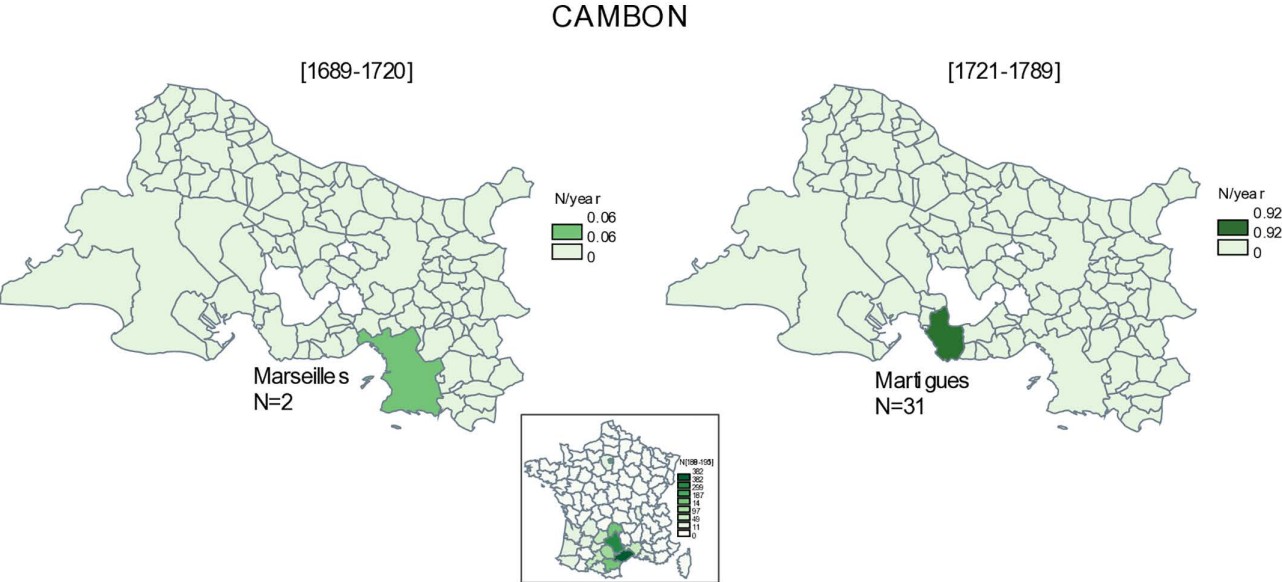

**Fig 2. Distribution of cambon/camboun surnames.** N is the number of births by municipality of the Bouches-du-Rhône department, between 1689-1720 and 1721-1789, and N/year is the number of births by year. The insert map is the distribution of CAMBON in French departments between 1891-1915. Original map created by the authors using the free software *Philcarto* (http://philcarto.free.fr), with base map © IGN – AdminExpress.

# LATAUD

[1689-1720]

[1721-1789]

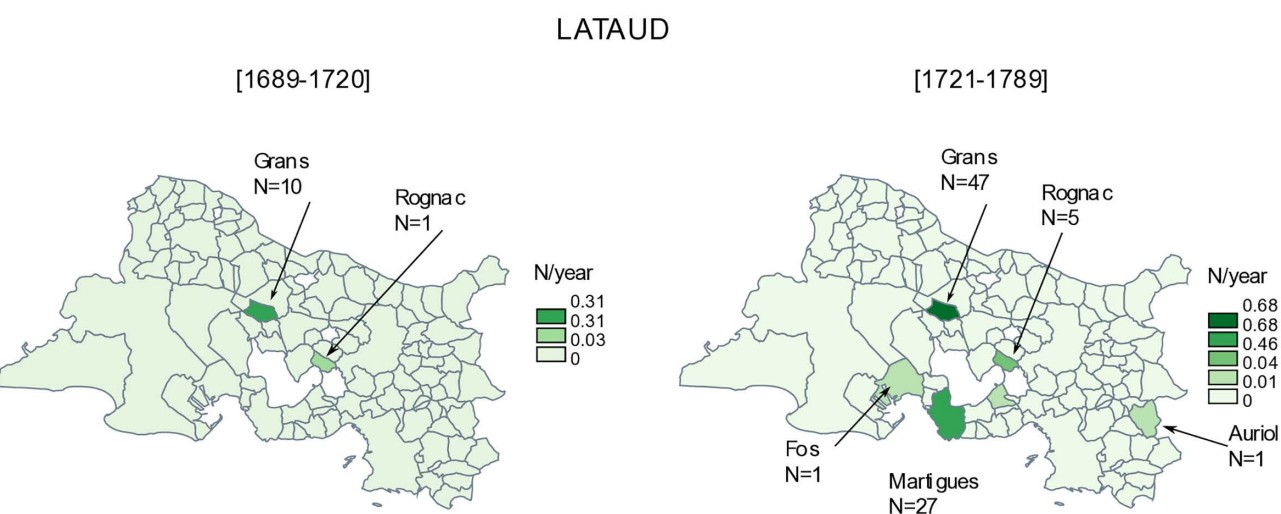

**Fig 3. Distribution of LATAUD surnames.** N is the number of births by municipality of the Bouches-du-Rhône department, between 1689-1720 and 1721-1789, and N/year is the number of births by year. The discretization method of the frequency is that of Jenks [29]. Original map created by the authors using the free software *Philcarto* (http://philcarto.free.fr), with base map © IGN – AdminExpress.

ANT(H)OINE, widespread throughout the study period? According to INSEE data, the ANTEAUME families were concentrated in the Gard department, in the municipality of Aramon, and the GANTEAUME families in La Ciotat.

Fig 4 shows the distribution of POUCEL/PONCEL, in Senas, but mainly in Aubagne before 1720, and in even higher numbers after 1720. The name is found in Martigues after this date (27 baptisms), which suggests that the POUCEL migrated to Martigues after the plague epidemic, where the fathers worked primarily as stonemasons.

The surnames CHALVE and CHALVI are present only after 1720, while CHAVE is common both before and after 1720. Does this mean that the spelling CHAVE changed to CHALVE just after and only after 1720? Or do these two names belong to two distinct genealogical lines?

The SOUR(R)EILLET surname is practically a geohapax in Martigues in the second period (1721–1789) as it is found neither in Bouches-du-Rhône in period P2 nor later in the INSEE file. However, the name could be a variant of SOULEILLET, which is a Type 3 surname in Martigues, common mainly in the Île district, while SOUREILLET is observed in the Ferrières district. According to the D-insee files, numerous SOULEILLET families lived in the Var.

### 4.4. Type 3

Type 3 names are those recorded in Martigues both before 1721 and after 1720. As such, they express the stable part of the population, which survived the plague without having to leave Martigues in large numbers and without being wiped out by the epidemic. Calculating, according to formula 1, the number of births of each of these surnames before and after the plague, weighted by the number of years covering the different periods (before 1721, 32 years; after, 69 years), gives an indication of the trend in the frequency of births for these names.

Table 5 provides the 30 names (with their variants) for which the number of baptisms increased the most between period P1 (1689–1720) before the plague and period P2 (1721–1789). These names, and their variants, include: TOURREL (1.95 to 4.15 births a year), FOUQUE, LAUGIER (0.67 to 2.07), AUDIBERT, CAMOIN, PIGNATEL and PHALIPON.

Table 5 also shows the 30 names, again with their variants, for which the number of births decreased the most between period P1 and period P2. These names include ARNAUD, VENEL, TOURRE, ROUSSIN and CAUDIERE.

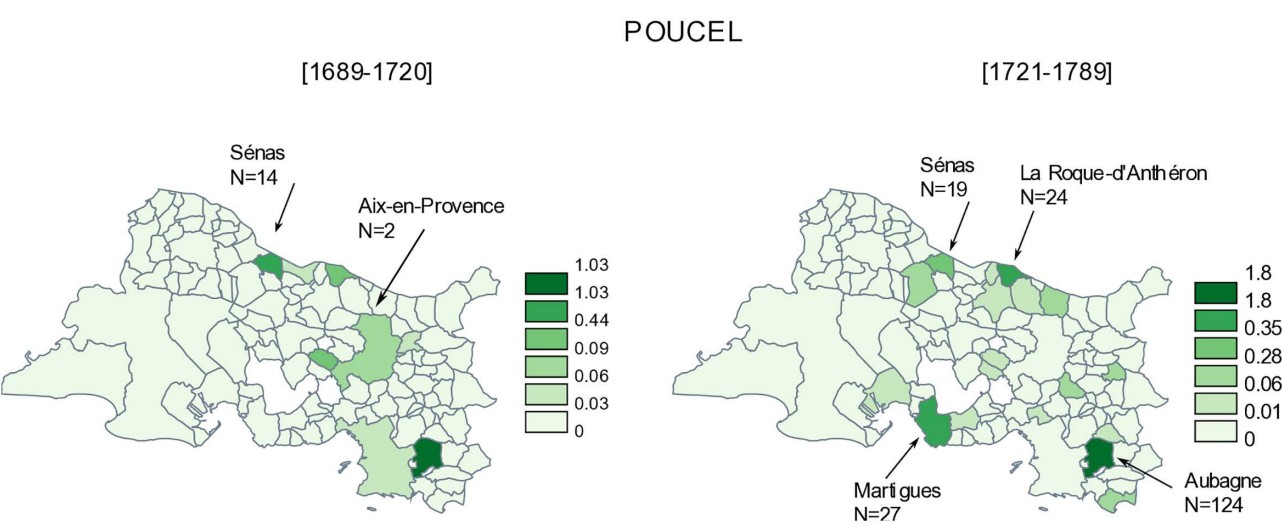

**Fig 4. Distribution ofpoucel/poncel surnames.** N is the number of births by municipality of the Bouches-du-Rhône department, between 1689-1720 and 1721-1789, and N/year is the number of births by year. The discretization method of the frequency is that of Jenks [29]. Original map created by the authors using the free software *Philcarto* (http://philcarto.free.fr), with base map © IGN – AdminExpress.

Fig 5 (top) illustrates the distribution of names ($N=236$) according to changes in the number of births per year between P2 and P1 (y-axis) and according to this number in P1 (x-axis): increase for values of over 0; otherwise, decrease. Fig 5 (bottom) shows the relationship between the number of births per year in P2 and in P1 (y-axis), according to the number in P1 (x-axis). This shows the names for which births were so low in the P1 period that they could also be considered as Type 2 names. This is true for PHALIPON, ROUBOU and ESPANET, among others.

The names TOURRE and TOURREL (and variants) are intriguing. It could be thought that these are two spelling variations of the same surname, TOURRE(L). But this does not appear to be the case. Both forms are found before and after 1720 and their reproductive trends are contrasted. TOURREL births were low before 1720 (0.68 per year) and increased strongly after (3.5), while TOURRE births averaged 2.8 per year before 1720 and just 0.88 after (see Fig 5).

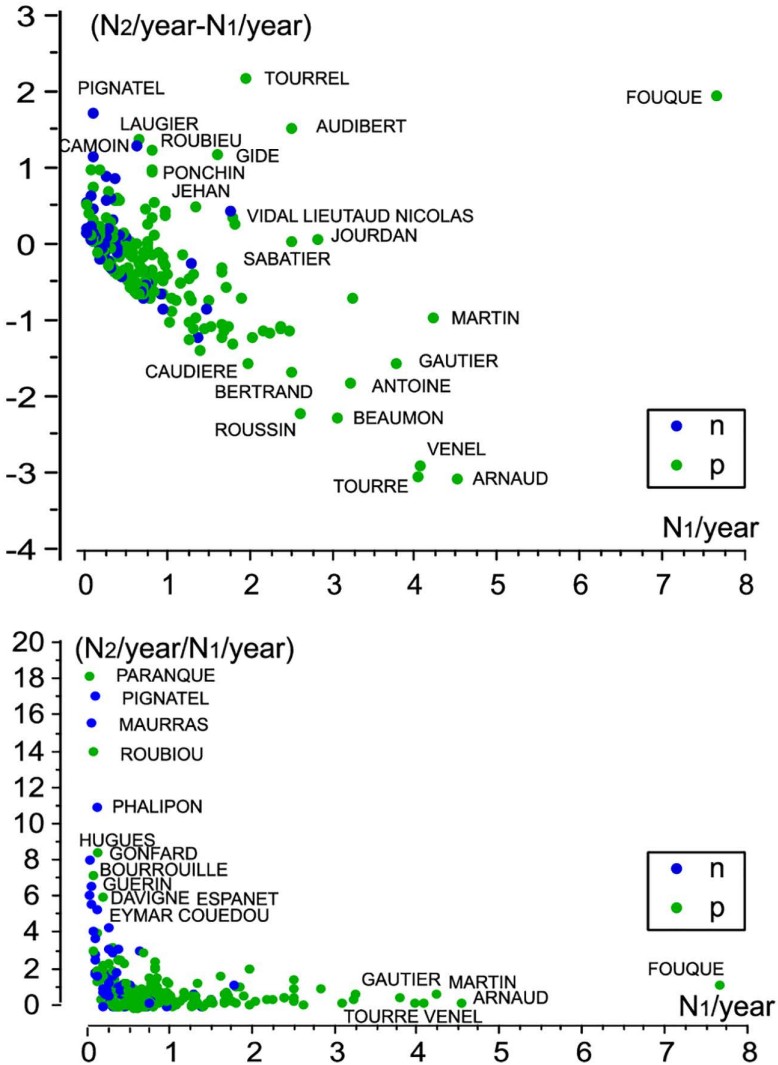

**Fig 5. Change in the number of births of Type 3.** Top: difference between the number of births per year N2/year during P2 (1721–1789) after the plague and the number of births N1/year during P1 (1689–1720) before the plague. Bottom: Ratio between these two numbers. p holds for names reported as those of people having died from the plague in the file of plague deaths, and n does not.

As seen earlier, PIGNATEL is a special case, as the form PIGNATEL without its variants was classified as a Type 2 name. In its 'pure' form, PIGNATEL is found rarely in the department before 1721, with four births in Allauch and three in the Aix-en-Provence area. Including all the variants of the surname, including those of PINATEL, this lemmatized surname becomes a Type 3 surname showing a substantial increase between the two periods, for example in Allauch, where these families were farmers and shopkeepers, and in La Ciotat and Gardanne.

The occurrence of the FOUQUE surname, already firmly established before the plague, with 6.2 births a year, rises sharply after the plague, with 8.8 births per year (Fig 5), even though the name is included among plague deaths.

Some names that were rare before the plague increased considerably after, including LAUGIER (0.5 births per year before and 1.9 after). Yet the LAUGIER are not listed among plague deaths. As this was a fairly common name in Bouches-du-Rhône, present in 51 out of 116 municipalities before 1720, two explanations are equally possible: a strong local increase in LAUGIER families after 1720 or their arrival from other municipalities after 1720.

A statistical analysis of the 236 Type 3 names shows in Table 6 that in period P1 the number of baptisms per year, N_1/year, was significantly higher among those whose name is included in the file of plague deaths ($p < 0.0001$), and that this number, N_2/year, is lower, non-significantly, in P2 ($p = 0.06$). The change in the number of births per year between the two periods is significant only for the names included in the file of plague deaths ($p = 0.0001$).

The higher number of Type 3 name births in P1 for those whose names are listed in the file of plague deaths (noted p) could result from a stronger and older presence in Martigues relative to the names not listed in the file (noted n) than those that are listed. The latter perhaps managed to escape the epidemic or were from a higher social class, which traditionally had fewer children. We may also conclude that the surnames listed in the D-Mdp file had significantly fewer children per year in the second period, as if there was a reduction in the number of children or a reduction in the name of reproducers, this regression not being observed among those not listed in the D-Mdp file. It is quite clear that differences exist in the reproductive behaviour of the people who endured the ordeal of the plague.

## 5. Discussion

The initial problem to be resolved in this work was the spelling variations of surnames. The transcription of surnames in the 17th and 18th centuries did not obey a strict spelling system. Names were sometimes written phonetically and sometimes feminized. The spelling of the surname of a single individual could change from one administrative document to another. And sometimes that change could be substantial. However, as reported by genealogists, the name of children did not differ from that of their father, as this is not observed in baptism certificates. Where a child was declared 'natural and legitimate', their surname was implicitly that of their father (for example, Jean Estaquier, son of Pierre, a sailor, and Jeanne Martin...'), ensuring a sort of lineal continuity with a certain degree of orthographic stability.

**Table 6. Averages (m) and standard deviations (s) in the number of baptisms per year.** $N_1$ represents the number of distinct names recorded between 1689 and 1720, while $N_2$ represents those recorded between 1721 and 1789. Names are further classified as present (p) or absent (n) based on their occurrence in the register of plague deaths (D-MDP). p represents probabilities associated with Student's t-tests.

| | | N_1 | N_2 | **Diff. |
|---|---|---|---|---|
| n (N = 54) | m | 0.375 | 0.479 | $p = 0.131$ |
| | s | 0.364 | 0.489 | |
| p (N = 176) | m | 1.017 | 0.738 | $p < 0.0001$ |
| | s | 1 | 0.987 | |
| *Diff | | $p < 0.0001$ | $p < 0.066$ | |

\* unpaired t-test; ** paired t-test.

To remain as faithful as possible to the historical source, as transcribed by genealogists, and to eliminate as far as possible the spelling disparities of a single surname, we opted for a dual approach, retaining all the possible forms of the names and agglomerating them in a consistent but refutable manner. The tables show the results of these two approaches. A strong correlation exists between them in that they both result in an almost identical list of surnames. For Type 1 names (Table 2), the correlation between the two options is $r=0.84$. The correlation is 0.93 for Type 2 names (excluding PIGNATEL and MAURAS/ MAURRAS) (Table 4) and exceeds $r=0.96$ for all Type 3 names. These strong correlations exist because there is practically always a dominant spelling, with numerous births, and minority spellings, sometimes simple hapaxes, which can have only a marginal influence. It can thus be considered that the aggregation of the number of the most frequent 'type' surname births with those of its variants, generally lower, does not lead to radically different results from a statistical standpoint, even if it turned out that some of these variants were grouped erroneously. But there are exceptions to this rule, including with the PIGNATEL/PINATEL and TOURRE/TOURREL names, as discussed earlier.

A further limitation is that of the quality of the sources. The data compiled by genealogists based on registered baptisms is often not entirely complete. We were obliged to reduce our analyses to the 1689–1789 period, which best covers all the baptisms transcribed in Martigues and in Bouches-du-Rhône. The data for prior periods are more incomplete and more difficult to use, owing to the poorer conservation of parish registers and palaeographic issues, which could explain the greater variations in the spellings of the names. We thought it preferable, then, to limit ourselves to this one-century period, even though this meant splitting the century (1689–1789) into two unequal periods on either side of the plague of 1720. Taking the example of the PONS surname, the 46 births before the plague (1689–1720) and the 46 births after (1721–1789) are not comparable because this represents more births per year in the first period (1.75) than in the second (0.70). This is why we standardized the number of births per year according to formula 1.

Owing to this temporal inequality, most of the births in the first period are essentially those due to parents born before 1689 and thus not included in the selected data sources. However, a person born at the start of the first period and thus included in the data will have children born mainly after 1710 and after the plague, if they survived it. But children born just before the plague, if they survived it, could complete their reproductive life, until 1789. Paradoxically, while the number of years in P1 is only half that in P2, the number of children per year is higher in P1, at 328 births per year, compared with just 252 in P2. So, there was a definite fall in fertility between the two periods, which cannot necessarily be attributed to the epidemic alone, though it may have played a role in the trend. Few historical demographic studies have focused on Provence in the eighteenth century. The works of Louis Henry [30] and Alain Blum [31] reveal marked regional differences in fertility patterns from the first half of the eighteenth century onwards, with the southeast showing lower completed fertility rates than northern and western France. In Martigues, fertility trends were likely intensified by the drastic decline in population. Following the plague, demographic recovery was neither rapid nor sustained, largely due to the stronger attraction of the neighbouring city of Marseille.

Regarding the number of surnames, the greater length of P2 (1721–1789) than P1 (1689–1720) gave more time for Type 2 surnames to take root in Martigues. This explains why the frequency of Type 3 names comes out at the same number for periods P1 and P2.

The way in which the two databases, on births (GENB-MAR) and plague victims (D-MDP), were established does not always enable us to know whether the simultaneous presence of the same name in the two databases corresponds systematically to the same people. This merely indicates that at least one person with that name died from the plague, suggesting that this may also have affected the rest of the families with the same name.

The surname method used here highlights the scale of the renewal of the population following the epidemic. Out of all the surnames prior to lemmatization (Table 1), 52% of those present between 1689 and 1720 (Type 1) disappeared between 1721 and 1789, while 37% of new names (Type 2) are reported after 1720. Only 21% of names (Type 3) are distributed throughout the entirety of the two periods. The same statistic established based on aggregated surname data, and solely including names for which at least four births are reported, shows that 23% of the names from the first period

disappeared, 26% of new surnames arrived after 1721, and 51% of the same surnames were observed before and after the plague. The difference between the two estimation methods lies in the fact that more hapaxes are observed before 1721, particularly feminized names, which are 'absorbed' by the second estimation method for which the variants of names are aggregated.

The results clearly show a significant renewal of the population after 1720—approximately 50%—possibly due to the plague outbreak. This development was also accompanied by a change in fertility, estimated here by the average number of children a year calculated per surname. This average falls significantly after 1720 ($p < 0.0001$) relative to its pre-1721 levels. This was noted earlier when discussing the contrast in the total number of births per year between the two periods. Was this caused exclusively by the plague? Or was it a generational trend? The first cause seems more likely, as the decrease was significant only for all the names present in the file of plague deaths, but not for those not included in the file and which may have been spared by the epidemic.

The plague also encouraged migratory movements, with new names, and thus new people, leaving or arriving in Martigues. Some names were no longer present in Martigues after the plague, and we are unable to determine where they went in the department. They account for 66% of the geohapaxes of Martigues before 1721 (Table 3). The destination of the remaining 33% can be traced, generally in the municipalities of the department, including those with the surname LATAUD from Grans, or in other departments, including those with the surname CAMBON. We have observed that the occupation of the fathers (sailor, farmer, 'day labourer') can be an explanatory factor in migratory behaviour (emigration and immigration alike) in Martigues before and after the plague.

The structure of the Martigues population, officially estimated at nearly 6,000 before the plague (see S1 File) and 7,258 in 1765, was thus profoundly disrupted by the plague of 1720, to the extent that the population was renewed by at least 50%, with turnover fuelled in part by the surrounding municipalities. Looking beyond the narratives that could be drawn from each individual story, the advantage of the surname method used here serves to paint a global picture of the demographic and anthroponymic disruptions generated by the dramatic episode of the plague epidemic in the historical context of the early 18th century.

## Supporting information

**S1 File. Sources.** S1.1. Parish of Jonquières (Municipal Archives of Martigues, GG 26-41), parish of Ferrières (Municipal Archives of Martigues, GG 46-52), parish of l'Île (Municipal Archives of Martigues, GG 11-22), subsidiary parish of La Couronne (Municipal Archives of Martigues, GG 43-45). S1.2. According to the 1716 enumeration conducted at the request of Intendant Cardin-Lebret, entitled *État du nombre des familles et des personnes de chaque lieu de Provence*, from which only the total population and the number of families for each locality in Provence have survived. This census is archived in the Bibliothèque Nationale under the reference, B.N. ms. fd. fr. 8908 and was partially published by Jean-Noël Biraben (1975, [2, t. 1, Annexe1]. This figure may be lower than the actual number, as it may take into account only the urban population and not that of the surrounding rural area. For comparison, the *Dénombrement général et particulier des communautés de la province et intendance de Provence, divisées par viguieries et fait par têtes au mois d'août 1765* records 5,559 inhabitants for Martigues, described as a "community of three parishes: L'Île, Jonquières, and Ferrières," and 1,519 in the countryside, giving a total of 7,119 inhabitants. To this must be added 100 Provençal foreigners and 39 non-Provençal foreigners. This document is transcribed in volume 5 (page 927) of Abbé Expilly's *Dictionnaire géographique, historique et politique des Gaules et de la France* (Paris, 1762-1770, 6 vol. in-fol. Unfinished). S1.3 Municipal Archives of Martigues, CC 390.
(DOCX)

**S2 Fig. Distribution of the number of baptisms by year (Y) in the four districts of Martigues between 1689 and 1789.**
(DOCX)

**S3 Fig. Table of available (1) or missing (0) data by year and district of Martigues.** Y:baptism year.
(DOCX)

**S4 File. Example of calculation (Formula (1)).** This example illustrates the calculation of the number of occurrences of surnames ABEILLE and ABILLE by period and by district, normalized according to the number of years of available observations in each district, and then combined to estimate the number of births per year and per period for Martigues as a whole. *N* denotes the number of births. The resulting estimates are reported in Table 5 in the main text.
(DOCX)

**S5 Fig. Distribution of the number N of births per year (N/year) between 1689 and 1720.** Jenks discretization method (29). Present administrative divisions. Original map created by the authors using the free software *Philcarto* (http://philcarto.free.fr), with base map © IGN – AdminExpress.
(DOCX)

**S6 File. Lemmatisation.**
(DOCX)

**S7 File. Table with the number of baptisms by surnames and districts, before the plague epidemic (1689–1720) and after (1721–1789), the list of surnames of people who died from the plague, and the number of births by name and by period in the districts of Bouches-du-Rhône.**
(XLSX)

## Acknowledgments

We would like to thank Geneanet for providing us with its data as part of a partnership agreement with INED and James Tovey for the English translation of an earlier version of the manuscript, and Christopher Leichtnam for this revised one.

## Author contributions

**Conceptualization:** Pierre Darlu, Isabelle Séguy.

**Data curation:** Pierre Darlu, Isabelle Séguy.

**Methodology:** Pierre Darlu, Isabelle Séguy.

**Writing – original draft:** Pierre Darlu, Isabelle Séguy.

**Writing – review & editing:** Isabelle Séguy.

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
