## [Decision Letter · Decision Letter 0]

17 Oct 2025

Dear Dr. Darlu,

Thank you for submitting your manuscript to PLOS ONE. After careful consideration, we feel that it has merit but does not fully meet PLOS ONE’s publication criteria as it currently stands.

**The reviewers found your work to be interesting and suitable for publication in PLOS ONE. However, they have identified several areas where improvements are needed. Therefore, we invite you to submit a revised version of the manuscript that addresses the points raised during the review process.. However, they have identified several areas where improvements are needed. Therefore, we invite you to submit a revised version of the manuscript that addresses the points raised during the review process.. However, they have identified several areas where improvements are needed. Therefore, we invite you to submit a revised version of the manuscript that addresses the points raised during the review process.. However, they have identified several areas where improvements are needed. Therefore, we invite you to submit a revised version of the manuscript that addresses the points raised during the review process**.

We look forward to receiving your revised manuscript.

Kind regards,

Grażyna Liczbińska

Academic Editor

PLOS ONE

Journal Requirements:

2. We note that your Data Availability Statement is currently as follows: The data required for this work has been jointly sent as an Excel spreadsheet.

3.  We note that Figure 1, 2, 3, 4 and S5 in your submission contain [map/satellite] images which may be copyrighted. All PLOS content is published under the Creative Commons Attribution License (CC BY 4.0), which means that the manuscript, images, and Supporting Information files will be freely available online, and any third party is permitted to access, download, copy, distribute, and use these materials in any way, even commercially, with proper attribution. For these reasons, we cannot publish previously copyrighted maps or satellite images created using proprietary data, such as Google software (Google Maps, Street View, and Earth). For more information, see our copyright guidelines: http://journals.plos.org/plosone/s/licenses-and-copyright.

1. You may seek permission from the original copyright holder of Figure 1, 2, 3, 4 and S5 to publish the content specifically under the CC BY 4.0 license.

**Additional Editor Comments:**

Thank you for your contribution. I encourage you to carefully address the reviewers’ remarks and resubmit your work.

Reviewers' comments:

Reviewer's Responses to Questions

**Comments to the Author**

1. Is the manuscript technically sound, and do the data support the conclusions?

Reviewer #1: Yes

Reviewer #2: Partly

Reviewer #3: Yes

Reviewer #4: Yes

2. Has the statistical analysis been performed appropriately and rigorously?

Reviewer #1: No

Reviewer #2: Yes

Reviewer #3: Yes

Reviewer #4: Yes

3. Have the authors made all data underlying the findings in their manuscript fully available?

Reviewer #1: Yes

Reviewer #2: No

Reviewer #3: Yes

Reviewer #4: Yes

4. Is the manuscript presented in an intelligible fashion and written in standard English?

Reviewer #1: Yes

Reviewer #2: No

Reviewer #3: Yes

Reviewer #4: Yes

Reviewer #1: The paper “The plague of 1720 and migration in Martigues (France) in the 17th and 18th centuries” explores the demographic impact of the 1720 plague on French-Mediterranean Martigues by comparing surname distributions in the decades before and after the outbreak. In principle, this is a good example for the usefulness of names-based, quantitative analyses of population turnover patterns in the absence of detailed migration statistics. In particular, I find the authors’ approach to form distinct “types” of surnames based on the pre/post-shock distributions quite creative and useful. However, based on several issues I outline below, I find that this article is currently not a good fit for PLOS One. My recommendation is to implement major revisions outline below. Alternatively, a more historically, possibly regionally specialized journal could be a better fit.

I have five major issues/recommendation with the current draft:

- Abstract and introduction should better highlight specific, interesting and new findings from the authors’ quantitative analysis. The current introduction makes quite general points about the usefulness of names-based analyses but does not clearly communicate what we learn from the application of these methods and sources to the particular 1720s Martigues outbreak. To be sure, some general conclusions are drawn in section 4 (“Discussion”), but I would like to see them highlighted from the beginning on. In short, I recommend that the authors give a clear idea, based on quantitative evidence, as to (1) “how important” out-migration, mortality and subsequent in-migration were for Martigue’s 1720 plague incident, (2) in how far their conclusion deviate or extend upon existing (qualitative) evidence on the 1720 plague, and (3) whether we would profit from attempting similar analyses for other epidemic outbreaks.

- Estimation of name frequencies (page 6): The authors present a pragmatic way to estimate true name frequency by period given that data for some year-region cells is missing in the original sources. However, from the description and formula given in the paper, it is unclear to me why the four different districts are not weighted by population size (or the number of births) when computing Martigues-wide surname indices. This seems sensible given that the 4 districts seem to vary strongly in population/birth number sizes (figure S3). If the authors think that such weighting is unnecessary in the aggregation, this choice should be explained.

- Interpretation of name-frequency changes as indicator for population turnover: It seems plausible to me that certain surnames would always be more likely to spontaneously reappear in a population, even if their original carriers all disappeared due to a shock (i.e. died or emigrated). I am thinking of surnames that relate to common occupations, places, resources or physical properties of the area. The authors’ current approach does not seem to account for this issue and rather conceptualizes the pool of possible names as static/given. If this is a defensible/useful assumption in the authors’ view, I would like to see a comment on this issue.

- Source of population inflow (line 710ff.): The authors claim that population inflow from surrounding municipalities could have accounted for a substantial part of the population replenishing after the plague shocks. I do not understand how this fits together with the earlier statement that many municipalities in the region were hit by the plague (not only Martigues), and 1/3 of the regional population died. Didn’t other towns in the region face similar population losses, and wouldn’t this undermine their ability to provide Martigues with new in-migrants?

- Exploring the correlation between name “types” and occupations: The original data used by the authors contains rich data on the occupations of inhabitants and the authors occasionally mention interesting concentrations of particular occupations among specific surnames or “types”. However, there is no attempt at a more formal analysis. I would like to see whether, statistically, surnames with an above-average representation in the three main socioeconomic classes in Martigues (seafaring/sea-related occupations, merchants, others such as agriculture) were more likely to disappear, newly appear, or stay stable between the two periods. This seems like an interesting and straight-forward extension to the paper.

There are some minor issues with the draft:

- Line 51: The statement that 242 communes in France were affected by the plague shock is not useful without an idea how many communes there were in total. Moreover, given the highly approximative death count (“nearly 120k”), it seems implausible that the count of affected communes is certain. I would omit this information and focus on the estimated death count in the region.

- Line 59: The sentence “See also the location of Marseilles” seems interrupted/incomplete (?).

- Line 86: Citation [7] does not seem to point to the right object in the bibliography.

- Line 101: Please clarify whether missing data is completely clustered by year (i.e. if a year is not missing, all baptisms in that year are transcribed). This seems to be assumed in your estimation, thus I would like to clarify.

- Line 109: What is a “significant” surname?

- Line 188: characters “1” and “[“ mixed up.

- Lines 312/313: You are referring to names with more than 3 baptisms, so the term “over-four” is misleading.

- Line 516: Character “]” missing.

- Line 666: You speculate that the remarkable fall in fertility after the plague “cannot necessarily be attributed to the epidemic alone, though it may have played a role in the trend.” At this point it is unclear to me on which basis you can make that claim. Starting in line 689, you then pick up this thought again, and seem to conclude that the plague indeed changed fertility behavior substantially? I recommend slightly restructuring/rewriting this part to minimize the need for interpretation on part of the reader.

- Figure S5 (appendix) shows many municipalities with 0 births per year between 1689 and 1720. Are these uninhabited? Seems implausible.

- The language is occasionally vague and colloquial (e.g. line 253: “(please excuse us for this gendered bias…)”).

Reviewer #2: The study investigates the impact of the plague of 1720 on the French town of Martigues. The authors compare the distributions of surnames, taken from local and regional parish registers before and after the deadly plague epidemic. The parish registers are accompanied by general information on surnames and death certificates. I believe that the study topic is interesting, not only to historical demographers, but the paper is not ready for publication in its current shape.

Methodologically, the authors use an interesting approach using surnames. My largest concern is the authors’ insufficiently critical discussion of the parish register data and their well-known limitations. Usually, parish registers from the early 18th century do not provide reliable information on the number of individuals present at a given moment in the parish. Individuals only appear in the data if they exhibit vital events like marriage, childbirth, and death. Labor migrants and other population groups are often recorded poorly or not at all. The estimation of the ‘population at risk’ is therefore biased and research questions addressing general trends of the study population must be assessed very carefully. I encourage the authors to think about the potential biases of using parish registers as a source. So far, the authors interpret their findings solely as a consequence of the plague epidemic and the resulting change in the population structure. Åkerman (1977) and Ruggles (1999) have dealt with the limitations of family reconstitution studies. Although this paper focuses on the distribution and development of surnames, the problems and limitations are potentially similar.

The authors mention on page 3 line 64 that there were 5,886 individuals living in Martigues in 1716. (Did this number derive from a census?) If their parish register data allows the reconstruction of life courses, it would be interesting to see how many individuals were present in Martigues in the same year according to the parish register data.

Another major flaw is the framing of the paper. It is unclear to me what the aim of the paper is. So far, the reader is educated in detail about the methods used, e.g. lemmatization techniques, and the results, changes in the surname distribution before and after the plague and the grouping into categories. But what do we learn from this? It remains unclear whether the authors want to test their methods or whether they have used the methods to investigate the change of the population structure in Martigues. The title suggests the latter, but for this enterprise the paper must focus and refer to other studies that have investigated population change in responses to disasters. On the last page of discussion, there are some speculations on the mechanisms of population change structure (plague mortality differentials between families, selective migration). Further, the whole paper lists only 14 references which only refer to data and methods issues. If the focus of the paper lies on the authors’ technique, then the framing and the title must be revised. It

Minor points

- There are no results mentioned in the abstract. The reader is just educated by the authors’ approach.

- Many abbreviations remain unexplained. For example, INSEE. I have a background in demographic research, and I know therefore that INSEE stands for the French ‘National Institute of Statistics and Economic Studies’, but not every reader knows that.

References

Åkerman, S., 1977. An evaluation of the family reconstitution technique. Scandinavian Economic History Review 25, 160–170. https://doi.org/10.1080/03585522.1977.10407879

Ruggles, S., 1999. The limitations of English family reconstitution: English population history from family reconstitution 1580-1837. Continuity and Change 105–130.

Reviewer #3: This article aims to analyse whether some families are more affected by the plague in 1720 Martigues than others, and how disruptive the plague was to a specific locality in terms of population dynamics and turnover, which can contribute to the debate on what the consequences of massive shocks such as major epidemics, for a population could be. I believe this is an interesting study, however, it is not embedded in the state of the art so far. I would have expected to find some references to historical/historical-demographical/demographic studies into the effects of major disruptive events such as wars and epidemics (Alfani on the plague for example, and/or Scheidel with The Great Leveler). It is important to add a bit of literature, to acknowledge the research that has been done so far, and to be able to put this finding into perspective.

The paper is furthermore rich in detail and shows the rigorous methodology of the authors, yet by explaining too many details and providing a too large number of examples, the narrative and the main message of the paper get lost to the reader. I would advise the authors to restructure the article and focus more on the main findings and trends, with fewer highly specific examples, details and exceptions.

For starters, the examples given from line 201 until 247 are a bit much. It would be better to provide an overview of how many names were difficult to lemmatize, and what percentage was easy, and sufficing with only one or two examples. The presentation of all these examples may appear thorough, but it is not clear to the reader whether this list is exhaustive. It is also not necessary to provide an exhaustive list, since the examples allow the reader to get an idea, and the results will later be given before and after lemmatization.

Moreover, tables 2, 4 and 5 are very elaborate, which hinders the quick conveyance of the main message to the reader. Could each table be moved to an appendix, and be replaced by more summarizing tables in the main text? I let it to the discretion of the authors to decide how to summarize and what message they want to convey, but a suggestion could be to summarize the cases by the Plague variable (where a p refers to plague deaths and an n to no plague deaths? This was not entirely clear, so this should be explained more carefully in the text as well. EDIT: it is only mentioned in line 603, which is a bit late). By structuring it in this way, it may become easier later on to say more about the percentages of names disappearing because of deaths, or migration. Another option would be to summarize the table by groups of how often a name appears (more applicable to Table 4, which I would advise to adjust in a similar manner as Table 2).

It may also be wise to order the table (the one that may be moved to an appendix) alphabetically (or by any other systematic approach, but I couldn’t find a system at the moment), to make it easier for your reader to digest.

Table 6: it is not clear to me what m and s refer to, could the authors specify? Overall, it may be helpful to include more informative headings in the tables, so they need less going back-and-forth between the explanatory text and the table.

The discussion section explains that the majority of the population disruption (names disappearing) is to be explained by plague deaths, and what happened in terms of migration after the plague. The discussion, however, fails to answer the initial objective listed in line 84-87 of studying whether some families were impacted more than others, based on socioeconomic characteristics. It is only mentioned in lines 704-706 that the occupation of the father could be an explanatory behaviour for migration, but a systematic analysis of socioeconomic background is lacking. I would advise the authors to either rephrase the aim of their study, or to include a systematic analysis of socioeconomic background (which might be a bit complex, but which would really add to the study!)

Reviewer #4: Although I consider the paper to be of high quality and likely acceptable for publication, I have a few minor comments and suggestions that the authors should address prior to final acceptance, which can be seen in the attached document.

.

Reviewer #1: **Yes:** Kalle KappnerKalle KappnerKalle KappnerKalle Kappner

Reviewer #2: No

Reviewer #3: No

Reviewer #4: No

---

## [Author Response · Author response to Decision Letter 1]

10 Dec 2025

This new version of our manuscript addresses the points raised by the reviwers and are iin the PONE-D-25-4689R1_Response to Rev-V2.docx already loaded

---

## [Decision Letter · Decision Letter 1]

23 Mar 2026

The plague of 1720 and migration in Martigues (France) in the 17th and 18th centuries

PONE-D-25-46489R1

Dear Dr. Darlu,

We’re pleased to inform you that your manuscript has been judged scientifically suitable for publication and will be formally accepted for publication once it meets all outstanding technical requirements.

Kind regards,

Grażyna Liczbińska

Academic Editor

PLOS One

Additional Editor Comments (optional):

**After reviewing the revised manuscript and your responses, I find that most comments regarding methodology, data presentation, literature, and clarity have been addressed. Where full compliance was not possible, you have provided convincing explanations, citing source limitations and historical context. While some concerns regarding interpretation and methodological choices remain, these do not undermine the validity of the results and have been reasonably addressed or justified.**

Reviewers' comments:

Reviewer's Responses to Questions

**Comments to the Author**

Reviewer #1: (No Response)

Reviewer #4: All comments have been addressed

2. Is the manuscript technically sound, and do the data support the conclusions?

Reviewer #1: Partly

Reviewer #4: Yes

3. Has the statistical analysis been performed appropriately and rigorously?

Reviewer #1: No

Reviewer #4: Yes

4. Have the authors made all data underlying the findings in their manuscript fully available?

Reviewer #1: Yes

Reviewer #4: Yes

5. Is the manuscript presented in an intelligible fashion and written in standard English?

Reviewer #1: Yes

Reviewer #4: Yes

Reviewer #1: I have looked through the authors revisions and I have mixed feelings about their attempts to increase the quality of the paper. Some of the low-hanging fruits like typos or missing explanations of details have been fixed. I also note that the introduction has been substantially improved, which was one of the most important issues I highlighted. Some more substantial conceptual issues have not been addressed too thoroughly, though:

- The authors declined to present substantial findings in the introduction, which-as a reader-I find odd, but can accept as a matter of stylistic choice.

- My question on the weighting of the four districts in the calculation of aggregate name frequencies was not really addressed. I am not convinced by the explanation of the authors and cannot help feeling that the average reader would still be confused about this.

- The authors argue that the fact that all affected locations in the region experienced emigration means that it is plausible that Martigue’s population was replenished by in-migration. I am still not convinced by that claim. If many people die in the region (120k according to the authors), it seems impossible that population stabilizes in every location through a migration channel as the total sum of available people diminished greatly. Why then would migrants stabilize the population in Martigues rather than other places?

- Several details on the handling of the data (like missing values for certain years) seem to be handled in a hand-waving manner; one of my main requests here was to clarify the assumptions and techniques used; as far as I can see, this has only been done rather selectively.

- My requests to tone down statements about the accuracy of highly speculative estimates like “120k deaths” was not met; rather, the authors seem strongly defensive about it. However, additional sources have been provided, it seems.

- As the authors stated that 242 communes in France were affected by the plague shock, I asked for a clarifying statement how many communes there were in total. The authors did not provide that; rather, it seems that the claim about the 242 communes has simply disappeared from the current draft without an explanation. In any case, the current version of the draft handles the discussion of the extent of the plague shock in a better way than before.

- My question about clusters in the missing data was not answered. To be clear, I asked whether the data in a given year Y is always either complete or completely missing; or whether there are years in which a fraction of the baptisms was available, but not all.

Let me stress that the paper is very readable in other aspects, and the core methodology (lemmatization) is surely an interesting contribution. Some requests have also been met in an excellent way, e.g. my question about the possibility of “spontaneous re-appearance” of certain names in the region.

Reviewer #4: The authors have carefully addressed the comments raised in my previous report, and the revisions have improved the clarity of the manuscript. I have no further concerns and believe the paper is suitable for publication.

.

Reviewer #1: No

Reviewer #4: No

---

## [Editor Report · Acceptance letter]

PONE-D-25-46489R1

PLOS One

Dear Dr. Darlu,

I'm pleased to inform you that your manuscript has been deemed suitable for publication in PLOS One. Congratulations! Your manuscript is now being handed over to our production team.

Kind regards,

on behalf of

Dr. Grażyna Liczbińska

Academic Editor

PLOS One